DOI: 10.1038/s41467-018-05844-8　　　**OPEN**

# Human neuroepithelial stem cell regional specificity enables spinal cord repair through a relay circuit

Maria Teresa Dell'Anno[1,2], Xingxing Wang [1,2], Marco Onorati [3,4], Mingfeng Li[4], Francesca Talpo[4], Yuichi Sekine [1,2], Shaojie Ma [4], Fuchen Liu [4], William B.J. Cafferty [2], Nenad Sestan [1,4,5] & Stephen M. Strittmatter [1,2,4]

Traumatic spinal cord injury results in persistent disability due to disconnection of surviving neural elements. Neural stem cell transplantation has been proposed as a therapeutic option, but optimal cell type and mechanistic aspects remain poorly defined. Here, we describe robust engraftment into lesioned immunodeficient mice of human neuroepithelial stem cells derived from the developing spinal cord and maintained in self-renewing adherent conditions for long periods. Extensive elongation of both graft and host axons occurs. Improved functional recovery after transplantation depends on neural relay function through the grafted neurons, requires the matching of neural identity to the anatomical site of injury, and is accompanied by expression of specific marker proteins. Thus, human neuroepithelial stem cells may provide an anatomically specific relay function for spinal cord injury recovery.

[1] Cellular Neuroscience, Neurodegeneration and Repair (CNNR) Program, Yale School of Medicine, New Haven, CT 06536, USA. [2] Department of Neurology, Yale School of Medicine, New Haven, CT 06520, USA. [3] Unit of Cell and Developmental Biology, Department of Biology, University of Pisa, Pisa 56127, Italy. [4] Department of Neuroscience, Yale School of Medicine, New Haven, CT 06520, USA. [5] Department of Genetics, of Psychiatry and of Comparative Medicine, and Yale Child Study Center, Yale School of Medicine, New Haven, CT 06510, USA. These authors contributed equally: Maria Teresa Dell'Anno, Xingxing Wang, Marco Onorati. Correspondence and requests for materials should be addressed to S.M.S. (email: stephen.strittmatter@yale.edu)

Traumatic spinal cord (SC) damage results in cell loss at the injury level, as well as disconnection of surviving neurons, with an irreversible interruption of the information flow to and from the brain. The implantation of neural stem cells (NSCs) at the lesion site has been considered an appealing potential treatment for decades, and several approaches have been proposed. Mechanistically, the hypothesized benefits of transplantation are diverse, including replacement of lost neurons, creation of a conducive axon growth environment for host axons, production of growth factors, and provision of glial cells to assist function of surviving neurons. In order for these mechanisms to occur, graft integration into the host is critical and defining the parameters that regulate its success is fundamental to facilitate translation of cell-based therapies to the clinic. Unfortunately, at present, neither the identity nor the selection path for the most appropriate cell population for optimal graft integration are known.

Human NSC transplants for spinal cord injury (SCI) have been limited to partially characterized human cell lines[1–3] or to fetal NSCs collected after 8 post-conceptional weeks (PCW)[4–6]. Although fetal NSCs can be propagated in vitro, neither their long-term stability nor the preservation of their regional identity in vivo have been demonstrated[7]. Fetal NSCs exhibit molecular markers suggestive of radial glia and appear to differentiate more easily toward the glial fate, whereas their neurogenic potential is largely restricted to GABAergic neurons both in vitro and in vivo[7,8]. In most previous reports, NSCs were cultured in suspension as neurospheres, a method that often leads to a significant reduction in self-renewal competency and in the neurogenic capacity of the cells[9,10].

As an alternative, human embryonic stem (ES) or induced pluripotent stem (iPS) cells are an in vitro source of neural progenitors and their application to SCI treatment is currently being investigated[11–14]. During human pluripotent stem cell differentiation, neural progenitors exhibit spontaneous self-organization into transient structures termed "rosettes". Cells within rosettes exhibit morphological and gene expression markers of neuroepithelial progenitors and are molecularly distinct from radial glia-like NSCs[15]. However, the identity and the physiological relevance of cells derived in vitro from pluripotent sources are unclear because cells could acquire transcriptional and epigenetic programs in vitro that diverge from cell states in vivo[16].

To understand how regional cell identity affects graft integration, we analyzed the engraftment of a novel human NSC population that retains over time the transcriptional profile acquired in vivo. In contrast to other NSC sources, human neuroepithelial stem (NES) cells derived from tissues collected at an embryonic stage of the neural tube development, typically from 5 to 8 PCW, possess unique advantages. NES cells can be propagated as monolayers for a virtually unlimited number of passages, retain a high and unaltered neurogenic potential over time and preserve the molecular and transcriptional signature of their tissue of origin[17,18].

We derived SC-NES cells from human post-mortem specimens and propagated them without genetic manipulation. Human SC-NES cells exhibited excellent integration properties in a rodent SCI model and established functional connections with local neurons. Through the application of chemogenetics to diverse behavioral paradigms, we show that SC-NES cells form a relay system through the lesioned area reconnecting spared host neural elements. In contrast, NES cells derived from neocortex (NCX-NES cells) fail to acquire a mature neuronal phenotype when transplanted into SC, fail to integrate and fail to extend neurites. Importantly, NCX-NES cell integration is dramatically enhanced in the cerebral cortex, demonstrating that anatomical matching of graft with recipient tissue is critical for functional neuronal networks. These findings provide key mechanistic, molecular and practical information to develop human cell transplantation therapy for SCI.

## Results

**Human SC-NES cells are tripotent and highly neurogenic.** Here we derived human SC-NES cells from six embryonic post-mortem specimens in a range of 5–8 PCW (Fig. 1a)[18]. The SC samples (Supplementary Fig. 1a) were dissected free of meninges and dorsal root ganglia and dissociated to a single-cell suspension. After 24 h of plating, SC-NES cells formed neural rosettes with typical radial organization and apico-basal orientation of the developing neural tube (Fig. 1a and Supplementary Fig. 1b). SC-NES cells exhibit progenitor cell characteristics and express the canonical NSC markers nestin, SOX2, vimentin, phospho-vimentin and PAX6 (Fig. 1b–d, f). KI67 staining confirmed the state of active proliferation of SC-NES cells in vitro (Supplementary Fig. 1c). Next, we investigated whether SC-NES cells retain their positional identity after long-term expansion. Therefore, we verified the expression of the caudal transcription factor HOXB9 (Fig. 1e) and confirmed immunolabelling specificity by lack of HOXB9 expression in NCX-NES cells (Supplementary Fig. 1d). Self-renewing SC-NES cells were maintained for >35 passages with no signs of senescence or chromosomal instability (Fig. 1g).

A 30-day differentiation protocol involving mitogen withdrawal coupled with neurotrophin addition induced the acquisition of a neuronal phenotype. SC-NES cells gave rise to RBFOX3- (also known as NeuN) positive neurons (Supplementary Fig. 1e) accounting for $79.2 \pm 0.1\%$ of cells (Supplementary Fig. 1i). Neurons also expressed the pan-neuronal markers MAP2, neurofilament (NEFL) and TUBB3 (Fig. 1h–i) whereas $11.2 \pm 0.2\%$ of cells were positive for the early post-mitotic neuronal marker doublecortin (DCX) (Supplementary Fig. 1f, i). Among mature neurons, 1% of the cells expressed choline acetyltransferase (CHAT), a feature of spinal motor neurons, in agreement with SC-NES cell anatomical origin (Fig. 1i Supplementary Fig. 1i). A subpopulation of cells expressed astrocyte and oligodendrocyte markers including GFAP and O4 (Supplementary Fig. 1g–i), confirming SC-NES cell tri-potent differentiation properties. In addition, we tested SC-NES-derived neurons for the expression of pre- and post-synaptic markers, PSD-95 and synaptophysin (SYP) (Fig. 1j, k), and for electrophysiological properties. Upon 90 days of in vitro differentiation SC-NES-derived neurons displayed ionic currents and elicit spontaneous repetitive action potentials (Fig. 1l, m), consistent with a fully functional neuronal phenotype (Supplementary Table 1).

**Human SC-NES cells survive and differentiate in a SCI model.** Human SC-NES cells were transduced with GFP-coding lentiviral vector and grafted into the lesion site of genetically immunodeficient mice 10 days after thoracic SC dorsal hemisection (Fig. 2a). Anatomical analysis revealed that SC-NES cells survived in all grafted animals and consistently filled the lesion cavity at 8 weeks post-grafting (Fig. 2b, c). Grafts in 3 out of 9 animals exhibited a rift near the graft center (Supplementary Fig. 2a) which contained collagen and SOX10-positive host Schwann cells (Supplementary Fig. 2b-c). The rift separated the graft into two segments and contained very few grafted-cell-derived neurites (Supplementary Fig. 2d).

In agreement with in vitro data, the majority ($55.6 \pm 3.5\%$) of transplanted SC-NES cells differentiated into RBFOX3-positive neurons (Fig. 2d–p), whereas $27.9 \pm 2.1\%$ of cells expressed

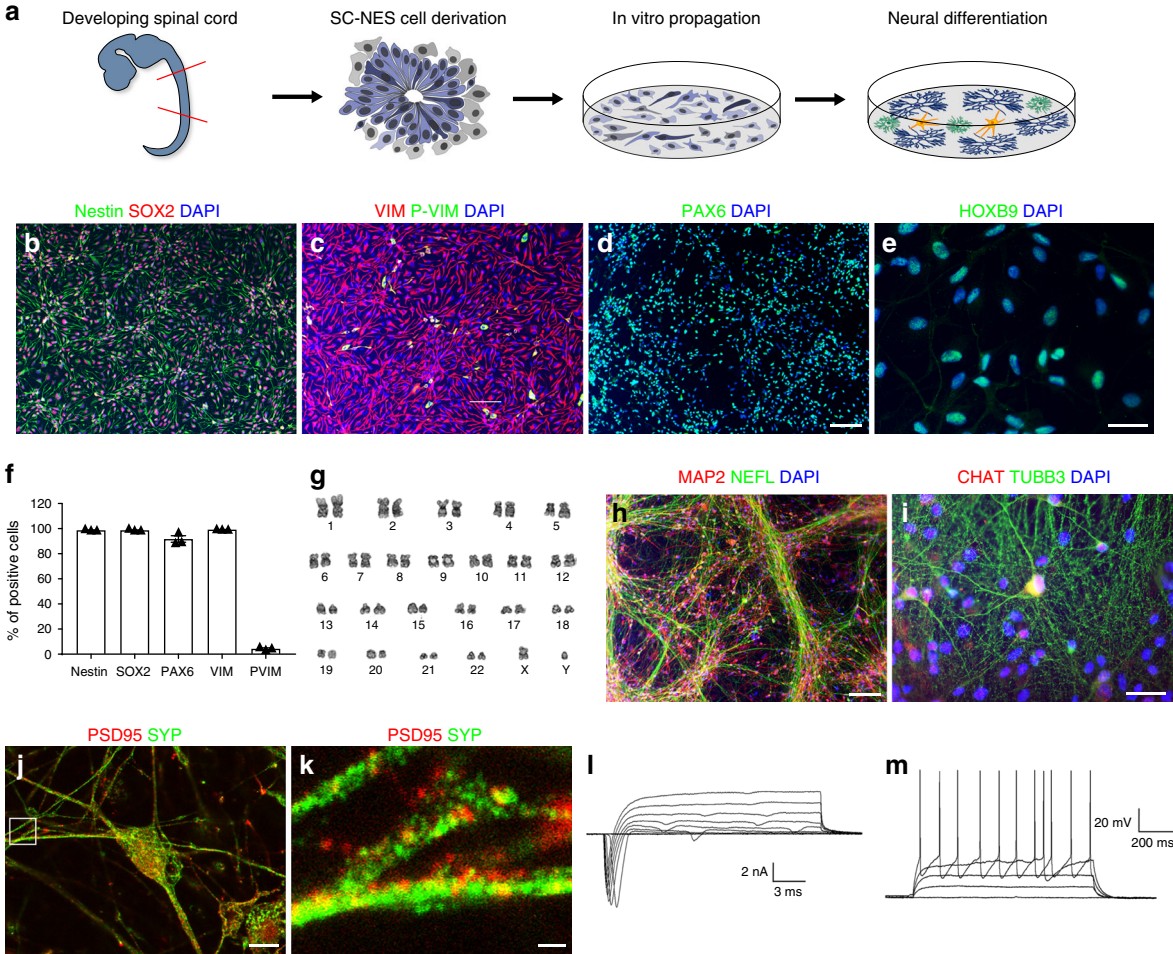

**Fig. 1** Characterization of human NES cells in proliferation and after in vitro differentiation. **a** Schematics of the experimental procedure. SC-NES cells were derived from the SC of a human post-mortem specimen at 6 post-conceptional weeks (PCW) and propagated in vitro. After differentiation SC-NES cells give rise to neurons, astrocytes and oligodendrocytes. **b–d** SC-NES cells are positive for pan-neural stem cell markers nestin, SOX2, vimentin (VIM), phospho-vimentin (P-VIM) and PAX6. **e** SC-NES cells retain their regional identity as proved by the positive staining for the SC-specific transcription factor HOXB9. **f** Quantification of nestin, SOX2, PAX6, VIM, P-VIM positive cells. **g** SC-NES cells retain a normal euploid karyotype after 25 passages. **h**, **i** Differentiation of SC-NES cells to MAP2-, TUBB3-, and neurofilament- (NEFL) positive neurons. SC-NES cells also generate choline acetyltransferase- (CHAT) positive neurons in agreement with their anatomical origin. **j**, **k** Mature neurons also express pre- and post-synaptic markers such as PSD95 and synaptophysin (SYP). **l** Total inward and outward ionic currents elicited at test potentials ranging from −70 to +40 mV from a holding voltage of −90 mV. (M) Subthreshold and suprathreshold voltage responses to a family of injected steps of current (0 pA; 20 pA; 40 pA; 60 pA; 80 pA) from a resting potential of −71 mV. Scale bars: **b–d** 100 μm; **e** 20 μm; **h** 100 μm; **m** 20 μm; **j** 10 μm; **k** 1 μm

human GFAP (Fig. 2e–p). APC-positive mature oligodendrocytes accounted for 1% of implanted cells (Fig. 2f). Co-staining for RBFOX3 and human nuclei confirmed the human origin of grafted neurons (Fig. 2g, h). These cells also expressed the mature neuronal proteins TUBB3 and HNCAM (Fig. 2i–j). In addition to general neuronal markers, 14.7 ± 1.4% of grafted SC-NES cells were positive for CHAT (Fig. 2k–l′). Inhibitory GAD1-positive neurons and excitatory VGLUT1 (SLC17A7)-positive neurons were less frequent, and accounted for 2% and 1% of grafted cells, respectively (Fig. 2m–o). No cells within the grafts expressed the serotonergic neuron marker, 5-HT.

In accordance with literature for other transplants[6,11,19,20], we observed that some grafted cells migrated into host SC and formed satellite clusters. We analyzed the SC from the brainstem to *conus medullaris*, and found that 5 out of 7 animals exhibited ectopic nodules. The vast majority of clusters were found just below the meninges (Supplementary Fig. 2e) within 11 mm from the graft center. Dispersion of cells into parenchyma occurred in one single recipient animal at a 100–200 μm distance (Supplementary Fig. 2f). In addition, some grafted cells were located in

the central canal (Supplementary Fig. 3c) at 7–12 mm from the injection site.

A subset of grafted cells (6.6 ± 1.7%) were positive for KI67 or expressed nestin and DCX, thus indicating that a fraction of cells remained in active phases of the cell cycle after 8 weeks (Supplementary Fig. 2g-i). However, the percentage of proliferating graft cells was not higher than the fraction of mitotic cells present in uninjured SC, which accounts for 10% of cells including central canal, gray and white matter[21]. Of note, SC-NES cells retained their regional identity in vivo and expressed the SC-specific transcription factor HOXB9 (Supplementary Fig. 2j-l).

**Human SC-NES cells elongate long distance axons in the host.** Graft-derived neurons extended many projections into host SC (Supplementary Fig. 3a). Emerging GFP-labeled processes were observed mostly in white matter (69.8 ± 3.5%) rather than in gray matter (30.1 ± 3.5%) (Fig. 3a, b), proving that SC-NES cells are not affected by myelin-associated inhibitors present in adult recipient tissues. We verified that SC-NES cell-derived neurons

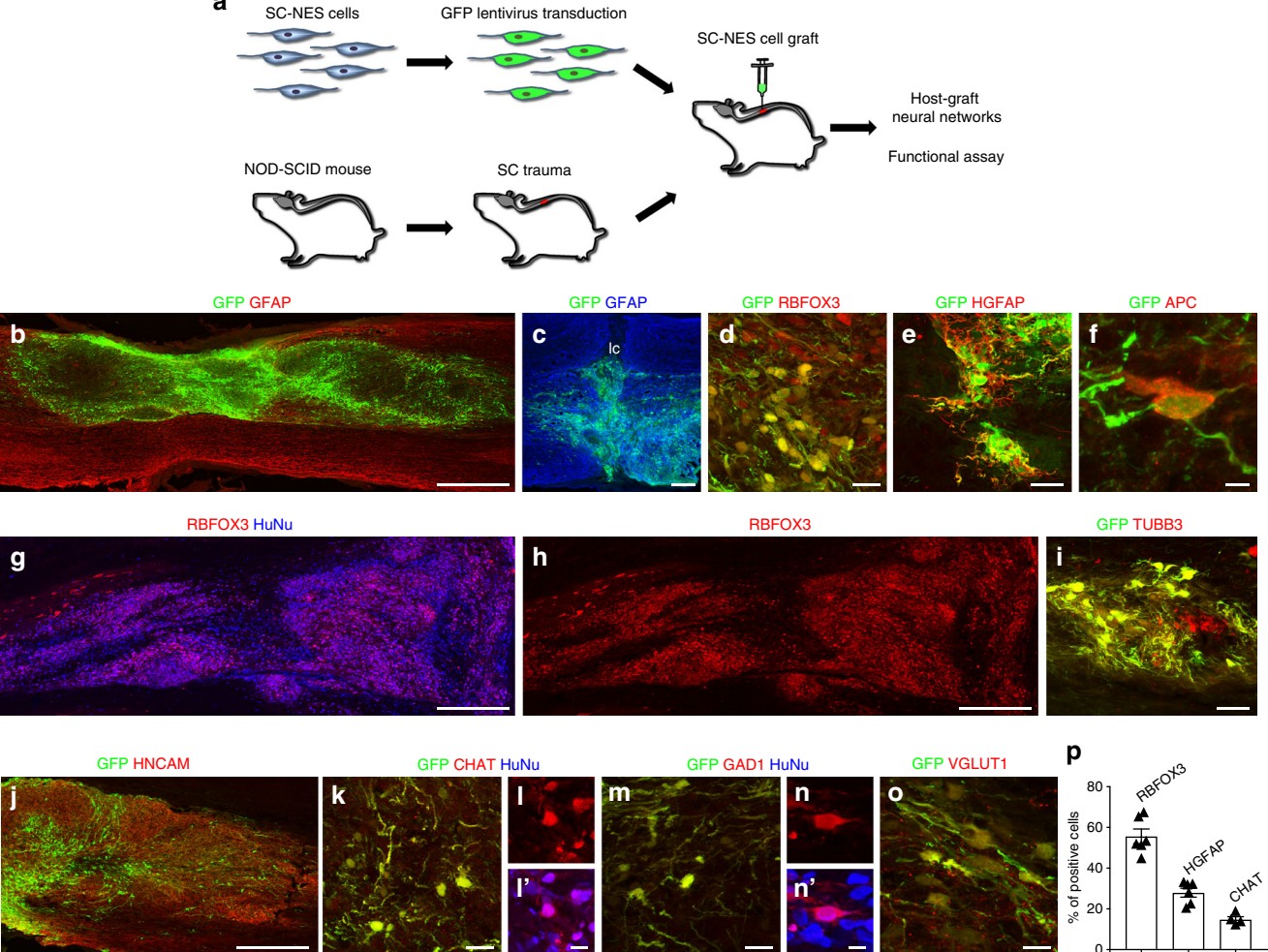

**Fig. 2** Human SC-NES cell graft survival and in vivo differentiation in a model of SCI. **a** Schematics of the experimental procedure. SC-NES cells were transduced with a GFP lentiviral construct before engraftment. Recipient immunodeficient (NOD/SCID) mice underwent dorsal SC hemisection and received SC-NES cell injection onto the lesion site 10 days after. **b** GFP-labeled SC-NES cells were implanted into the site of the injury. Horizontal section immunolabelled for GFP and GFAP indicates that cell implants survive and distribute into the lesion cavity. Rostral is to left, caudal is to right. **c** High magnification of the lesion core (lc) showing the correct placement of the cells in the injured area. **d** The majority of the cells within the graft differentiates toward the neuronal fate giving rise to RBFOX3-positive neurons. **e**, **f** SC-NES cells also generate astrocytes positive for human GFAP (HGFAP) and oligodendrocytes positive for APC. **g**, **h** Low magnification image of grafted cells immunostained with RBFOX3 and human nuclei (HuNu) antibodies, confirming the human origin of differentiated cells. **i**, **j** Grafted GPF-labeled cells are also positive for the neuronal markers TUBB3 and HNCAM. **k**–**l′** A fraction of GFP-labeled SC-NES cells colocalize with the motor neuronal marker choline acetyltransferase (CHAT) and co-staining with HuNu antibody confirms the human origin of the cells. **m**–**n′** Immunostaining for GFP, HuNu and the inhibitory neuronal marker glutamate decarboxylase 1 (GAD1) showing colocalization of GAD1 staining with GFP and HuNu. **o** Double labeling for GFP and vesicular glutamate transporter 1 (VGLUT1) showing a graft-derived human neurite co-expressing VGLUT1. **p** Quantification of RBFOX3-positive neurons (55.6 ± 3.6%) and HGFAP-positive astrocytes (27.9 ± 2.2%) representing the two largest cellular subtypes in the graft. Among neurons, CHAT-positive motor neurons were the most abundant subtype accounting for up to 14.7% of grafted cells. Scale bars: **b** 500 μm; **c** 200 μm; **d** 20 μm; **e** 40 μm; **f** 4 μm; **g**, **h** 500 μm; **i** 40 μm; **j** 250 μm; **k** 20 μm; **l**, **l′** 10 μm; **m**, 20 μm; **n**, **n′** 10 μm; **o**, 20 μm

do not express Nogo receptor 1 (NgR1), which is known to mediate axonal growth repression exerted by ligands Nogo A, Myelin Associated Glycoprotein (MAG) and Oligodendrocyte Myelin Glycoprotein (OMgp)[22–25], thus providing a possible mechanism for the robust axonal growth observed in white matter (Supplementary Fig. 3b). Notably, GFP-positive projections grew in parallel linear trajectories in white matter and were ramified in gray matter within close proximity to host neurons in a pattern that is very similar to endogenous axonal fibers (Fig. 3c–e). The axonal nature of projections was confirmed by GFP colocalization with the axon-specific marker, human neurofilament (NEFL) (Fig. 3f, g). SC-NES cell-derived axons did not

appear to be myelinated by host oligodendrocytes (Fig. 3h). The absence of myelination of human axonal fibers in rodent hosts has also been reported by others[11] and may be attributable to a lack of inter-species recognition.

Human SC-NES cell-derived axons extended over the entire length of the SC from the injection site in the thoracic segment. GFP-labeled terminals were observed cervically up to the *medulla oblongata* and caudally in the *conus medullaris* (Fig. 3i–l). The distance traversed by GFP-positive axons was 13 spinal segments rostral and 6 segments caudal from the implantation site with a length of 4 cm. The GFP-positive axons found in host tissue emerged directly from the main body of the graft and not from

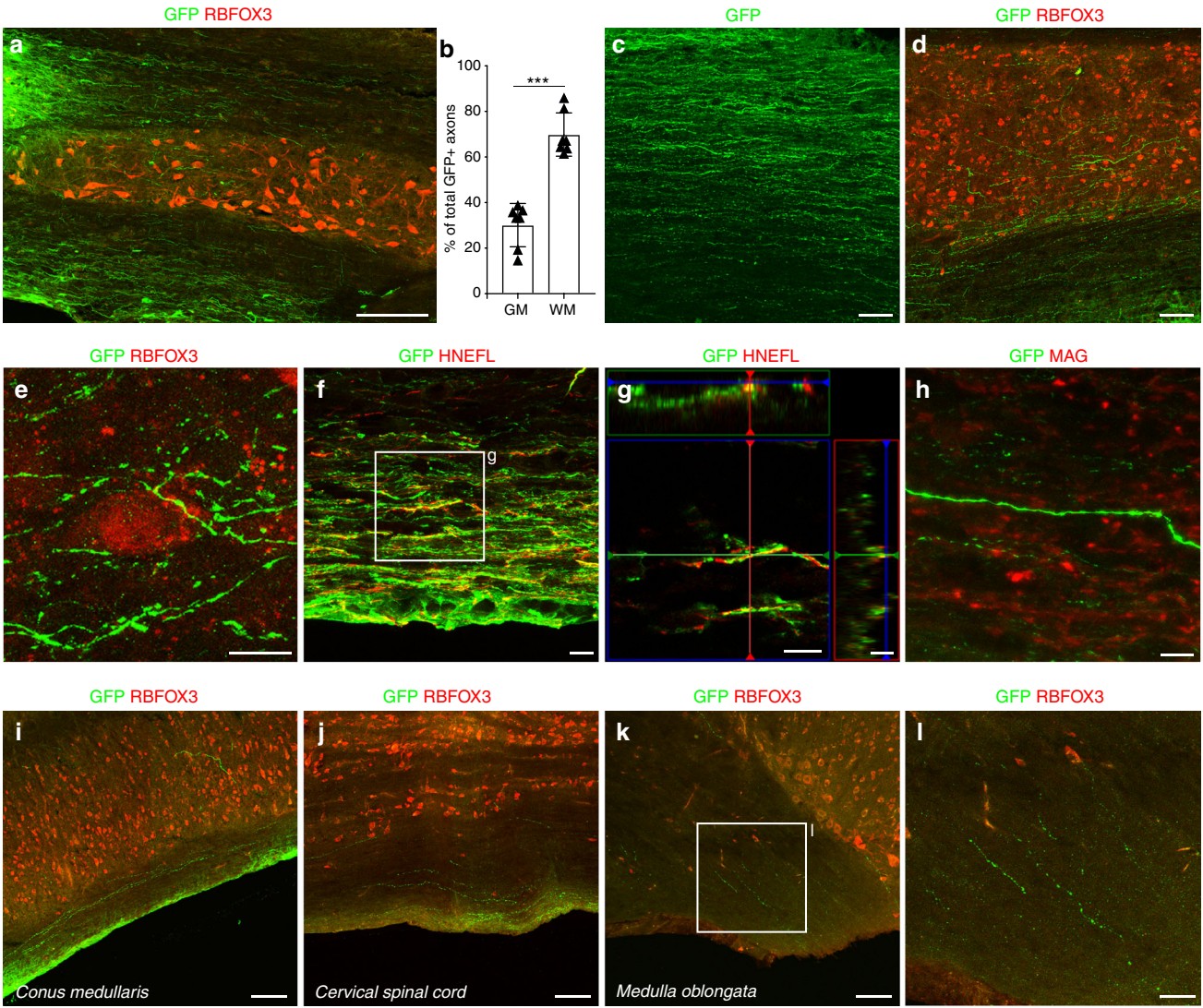

**Fig. 3** Extensive axonal outgrowth from human SC-NES cell grafts. **a** GFP and RBFOX3 immunolabeling on a horizontal section of SC grafted with SC-NES cells. GFP-expressing cells show a massive axonal elongation rostral and caudal to the trauma site (caudal shown). **b** Quantification of graft-derived axons in the gray (GM) and white matter (WM) of the recipient SC showing a preferential axonal growth in the WM. **c, d** GFP-positive axons grow in a parallel pattern in the WM and acquire a more ramified morphology in the GM identified by the RBFOX3 staining. **e** GFP-expressing axon terminals are closely associated with host RBFOX3-positive neurons. **f, g** GFP-labeled projections arising from the graft express human neurofilament (HNEFL), confirming their identity as axons. **h** GFP and myelin associated glycoprotein (MAG) staining reveals a lack of myelination of the human fibers. **i–l** GFP and RBFOX3 staining of the recipient SC at different levels. Human GFP-positive axonal fibers elongate throughout the entire length of the SC from the original injection site in the thoracic segment. GFP-positive axons can be found in the cervical portion and more rostrally in the *medulla oblongata*. Human axons also grow caudally reaching the *conus medullaris*. Data are expressed as mean ± s.e.m (***$P < 0.001$; Student's $t$-test). Scale bars: **a** 200 μm; **c, d** 50 μm; **e** 10 μm; **f–h** 10 μm; **i–k** 100 μm; **l** 50 μm

the few cells of ectopic central canal nodules, which appeared tightly clustered and devoid of protrusions >10 μm (Supplementary Fig. 3d-g).

Thus, human SC-NES cells exhibit excellent survival properties and appear impervious to myelin-associated inhibitors with consequent growth of long-distance projections to re-establish a connection between the SC segments above and below the trauma.

**Human SC-NES cells establish a functional neuronal network.** We sought to determine whether SC-NES cells form neuronal networks within the host. To this purpose we labeled grafts with anti-human synaptophysin (HSYP) and examined RBFOX3-positive neurons in the gray matter of recipient SC (Fig. 4a). We found HSYP-positive puncta surrounding host neurons,

consistent with synapses between grafted cells and the host (Fig. 4b). Similarly, we investigated the formation of synaptic connections between host fibers and the graft. In particular, we analyzed two descending pathways: the corticospinal tract (CST) and the raphespinal tract. Biotin dextran amine (BDA)-labeled CST fibers regenerated into the graft (Fig. 4c, d) and colocalized with the pre-synaptic marker SYP, as well as the GFP-positive human projections, thus establishing a mechanism for host-graft connectivity (Fig. 4e). No CST axons extended beyond the graft into the caudal SC. Raphespinal 5-HT-positive fibers were also found to grow into the graft (Fig. 4f, g) and formed synapses with human cells (Fig. 4h). Notably, 5-HT fibers were observed throughout the entire graft (Fig. 4j–l).

Grafted animals were also subject to functional analysis. Behavioral outcomes of control group and SC-NES cell recipients

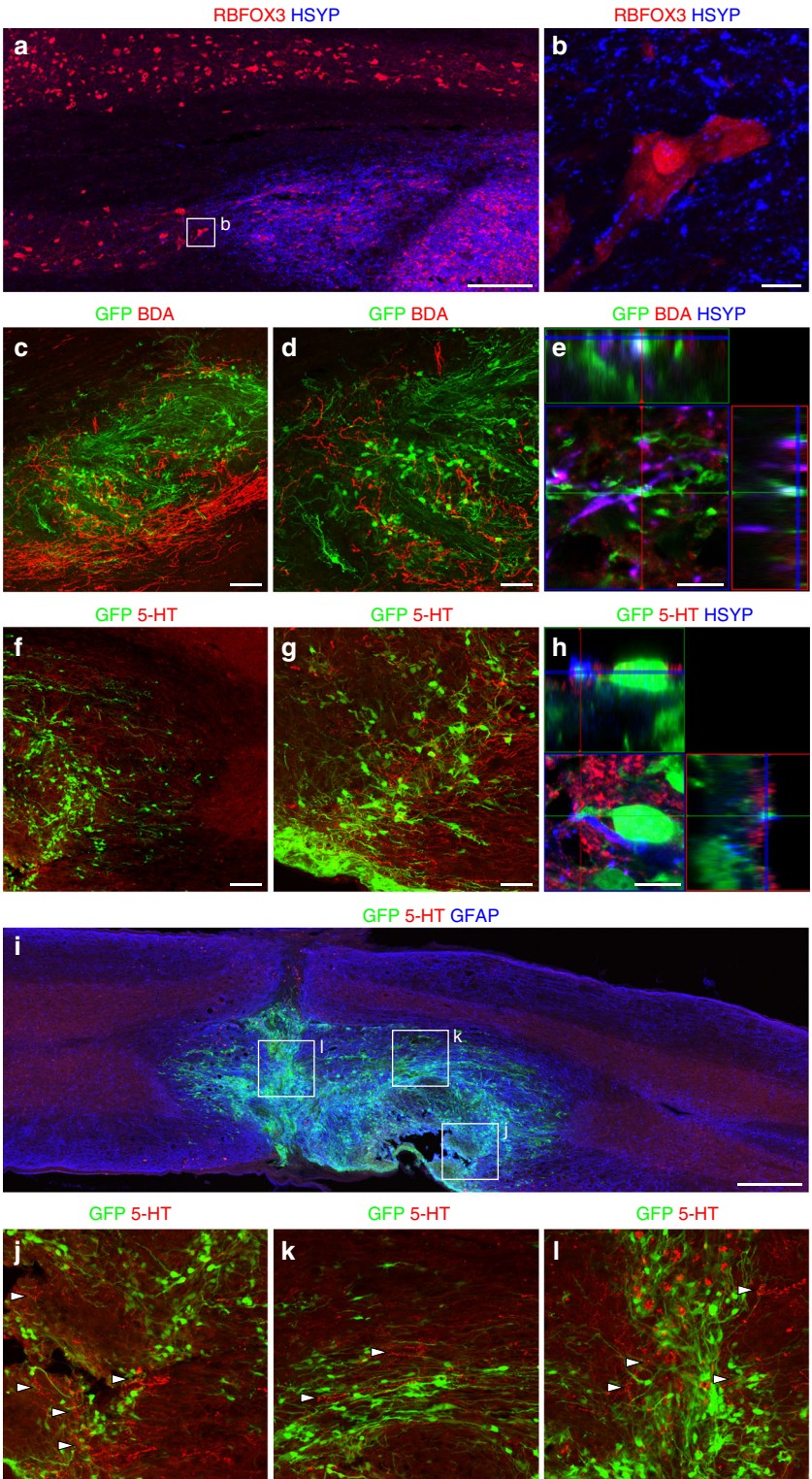

**Fig. 4** Graft-Host and Host-graft interactions. **a** Human synaptophysin (HSYP) staining identifies grafted SC-NES cells. RBFOX3-positive neurons indicate the gray matter of the recipient SC. **b** High magnification image of a host RBFOX3-positive neuron surrounded by HSYP-positive puncta, suggesting the establishment of synaptic contacts between the graft and the host. **c** Host corticospinal axons labeled with BDA regenerate into GFP-expressing SC-NES cell grafts. **d** Higher magnification of BDA positive fibers in the graft. **e** Adjacent signal of GFP, BDA and HSYP signal indicating the formation of novel synapses between the host CST and implanted cells. **f** Host serotonergic axons immunolabelled for 5-HT elongate into the GFP-positive SC-NES cell graft. **g** Higher magnification of 5-HT positive fibers in the graft. **h** Host 5-HT fibers are proximal with HSYN and GFP-positive terminals indicating the establishment of synaptic contacts between the host raphespinal fibers and grafted cells. **i** 5-HT-positive fibers elongate through the entire length of the cell implant. Rostral is on the right and caudal is on the left. **j–l** High magnification of boxed areas in (**i**) showing serotonergic terminals intermingled with GFP-labeled graft cells close to the rostral graft-host interface (**j**), in the middle of the graft (**k**) and in proximity of the lesion core (**l**). Arrowheads indicate 5-HT-positive fibers. Scale bars: **a** 200 μm; **b** 10 μm; **c** 100 μm; **d** 50 μm; **e** 10 μm; **f** 100 μm; **g** 50 μm; **h** 10 μm; **i** 500 μm, **j–l** 50 μm

were measured by Basso Mouse Scale (BMS) starting 7 days after injury, but prior to engraftment[26]. At the pre-graft evaluation, both groups had substantial loss of locomotor function with an average BMS score <1 (slight ankle movement). The SC-NES cell recipient animals showed a mild improvement by weeks 7 and 8 with a final score of 4 (occasional stepping), whereas control

animals scored 3 (paw placement with or without weight support) (Fig. 5a). We determined the extent of tissue spared from injury, and confirmed that the lesion was comparable between controls and SC-NES grafted mice (Fig. 5b). Next, we performed a correlation analysis between spared tissue and BMS score at 8 weeks. We observed a statistically significant difference in the

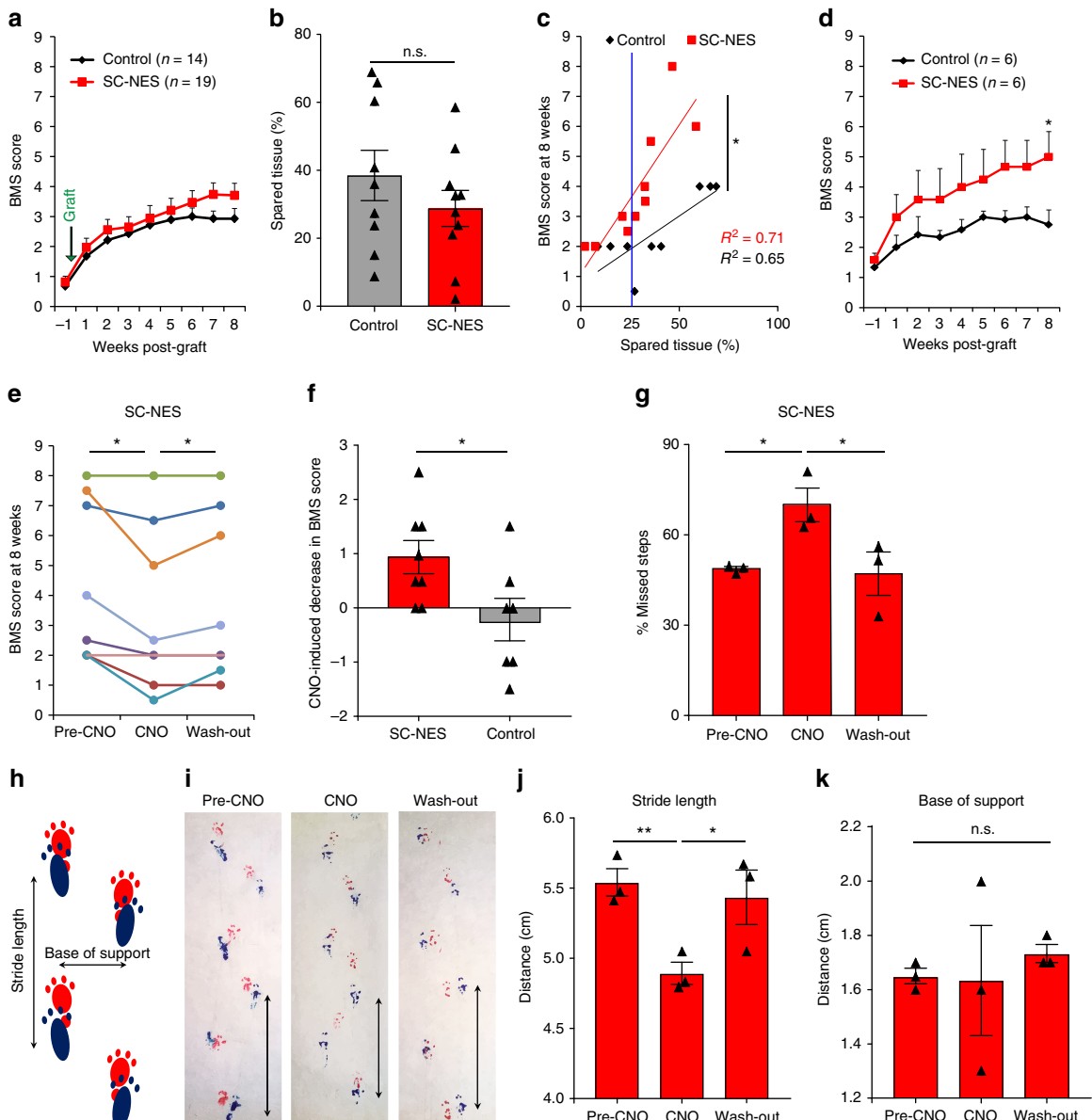

**Fig. 5** Human SC-NES cells form a relay system in the injured SC. **a** Hindlimb locomotion: BMS scores before and after dorsal SC hemisection in animals subject to cell implant ($n = 19$) and in controls ($n = 14$). The first time point indicates the locomotor performance one week after injury and three days prior to cell engraftment. **b** Spared tissue quantification in control and treated animals. **c** Correlation analysis of spared tissue percentage and the BMS score at 8 weeks in a subgroup of animals. Each symbol represents an animal. The correlation shows a significant difference between SC-NES cell recipient animals and the control group. The blue line indicates the percentage of spared tissue equal to 25%. Animals on the right of the blue line were selected for a stratification of the original cohort of animals shown in **a**. **d** BMS scores including SC-NES cells recipient animals ($n = 6$) and control animals ($n = 6$) with a percentage of spared tissue greater than 25%. **e** BMS score of SC-NES cells recipient animals at 8 weeks. Each dot represents an animal. SC-NES cells were transduced with the hM4Di-mcherry construct prior to surgical implantation. Animals were scored before CNO administration (Pre-CNO), 30 min after CNO administration (CNO), and 24 h after (wash-out). **f** The per-animal change in BMS score before and 30 min after CNO administration is plotted for SC-NES cell-treated and control animals. **g** Grid-walking test: percentage of missed steps of SC-NES cell recipient animals before CNO treatment, under CNO effect and 24 h after drug administration (wash-out). **h** Schematics of the foot print test. Base of support and stride length were measured. **i** Representative images of SC-NES cell recipient animals footprints before, during and after CNO injection. Black arrows indicate the stride length for one representative step in all three conditions. **j** Quantification of stride length in SC-NES cell recipient animals pre-CNO, upon CNO treatment and after drug wash-out. **k** Quantification of base of support distance. Data are expressed as means ± s.e.m. (*$P < 0.05$; **$P < 0.01$; Student's $t$-test for comparisons between two groups and repeated measures ANOVA for BMS tests)

linear regression between the two groups indicating that cell recipient mice had better locomotor performance than control animals with a comparable injury (Fig. 5c). The correlation analysis revealed that the divergence in the behavioral outcome was most prominent for animals with mild to moderate injury, defined as spared tissue >25% (Fig. 5c). Therefore, we stratified the behavioral data, including mice with a lesion not greater than the 75%. For mice with such mild to moderate injuries, there was significantly greater performance by cell recipient animals with a BMS score of 5 (frequent or consistent plantar stepping) by week 8 post-implant, as compared to similarly injured control animals scoring <3 (Fig. 5d).

The beneficial effect of neural grafts within the injured SC has been associated with diverse mechanisms. However, at present, no direct evidence has been provided to demonstrate that neural grafts can generate novel circuits within recipient SC. Here, to assess the necessity for SC-NES cell neural relay function, we transiently silenced their electrical activity using the hM4Di-DREADD receptor. hM4Di is an engineered human muscarinic receptor designed to be responsive to the otherwise inert ligand Clozapine-N-Oxide (CNO) and mediates neuronal membrane hyperpolarization with the consequent reduction in action potential firing[27,28]. First, we assessed the efficacy of hM4Di-CNO for SC-NES cells in vitro. We differentiated SC-NES cells to achieve firing of spontaneous action potentials, then, we transduced SC-NES cell-derived neurons with an hM4Di-mcherry lentiviral construct (Supplementary Fig. 4a). Application of CNO caused significant hyperpolarization of membrane potential and decrease in action potential frequency (Supplementary Fig. 4b-d). SC-NES cells expressing hM4Di-mcherry were then transplanted into hemisected SCs of immunodeficient mice. Animals were assessed by BMS scale at 8 weeks post-implantation before CNO administration (Pre-CNO), 30 min after injection (CNO) and 24 h later (wash-out). Upon CNO administration, 6 out of 8 animals showed a significant decline in locomotor skills, with restoration of function after CNO wash-out (Fig. 5e, f). Control lesioned animals underwent identical functional testing but did not exhibit any significant variation in their performance with CNO treatment (Fig. 5f).

The same cohort of grafted and control mice was also evaluated by grid-walking test where animals are assessed for their ability to walk on a suspended grid by quantifying the percentage of incorrectly placed steps[29]. SC-NES cell grafted animals showed a significantly greater percentage of missed steps during CNO treatment as compared to pre-CNO and post-CNO wash-out performances (Fig. 5g). For control animals there was no detectable effect of drug on the grid-walking competency (Supplementary Fig. 4e).

Finally, gait was assessed by the foot-print test[29]. We measured the distance between the hindlimbs (base of support) and the stride length (Fig. 5h). Paired testing revealed that, compared to the pre-CNO assessment, drug treatment resulted in a significant reduction of the stride length for SC-NES cell recipient animals with a recovery after wash-out (Fig. 5i–j). CNO did not alter control animals' performance (Supplementary Fig. 4f). Base of support was not changed by CNO administration in either group (Fig. 5k).

These data show that the transient silencing of SC-NES cells abrogates the beneficial effect of transplantation in three different functional paradigms. The temporary interruption of SC-NES cell electrical activity through a pharmaco-genetic tool has a direct impact on the functional outcome of grafted mice thus providing evidence that while SC-NES cells can act as a scaffold for severed axons to regenerate, they can also create a relay system essential for functionally reconnecting supraspinal axons with denervated target neurons below the level of the trauma.

**Human NES cell regional origin effects graft integration**. Next, we explored whether the beneficial effects of SC-NES cells were related to the regional identity of the cells. To this purpose, we compared human SC-NES cells with human NES cells derived from the NCX of an age-matched embryo. Upon in vitro differentiation, SC- and NCX-NES cells appeared indistinguishable: both displayed a mature neuronal phenotype and gave rise to a similar number of RBFOX3-positive neurons (Supplementary Fig. 5a-c)[18]. However, when transplanted in the injured SC, NCX-NES cells did not integrate in the parenchyma and formed dense clusters at the injection site with drastically reduced axonal elongation compared to their SC-derived counterparts (Fig. 6a–c). Notably, SC-NES cells distributed more broadly into SC tissue over a much larger area (Fig. 6d–f). The significant reduction in the volume of NCX-NES cell grafts did not appear to be associated with a higher number of apoptotic events as no statistical difference was observed in the quantification of active caspase 3 positive-cells in both SC- and NCX-NES cell implants (Supplementary Fig. 5d). In addition, KI67 staining performed at an early stage of in vivo cell differentiation (i.e., 2 weeks post-implantation) did not reveal a significant difference between the two cell types (Supplementary Fig. 5e), thus excluding enhanced mitosis as a possible mechanism behind the larger volume of SC-NES cell transplants. NCX-NES cells in vivo also appeared to be substantially undifferentiated. Immunostaining for pan-neuronal markers RBFOX3 and TUBB3 revealed no colocalization with NCX-NES cells (Fig. 6g–h), whereas the vast majority of the cells remained immuno-positive for the neural progenitor marker nestin (Supplementary Fig. 5f-g).

We analyzed interactions between NCX grafts and the host, but found no BDA-labeled host CST fibers establishing connections with grafted NCX-NES cells (Supplementary Fig. 5h). Unlike SC-derived NES cells, NCX-NES cells failed to produce any functional improvement in recipient animals over 8 weeks (Fig. 6i and Supplementary Fig. 5i). This cannot not be attributed to greater injury severity, since spared tissue was similar in graft recipients and controls (Fig. 6j).

Of note, when placed into the motor cortex of immunodepressed mice, NCX-NES cells displayed a mature morphology at 8 weeks after engraftment with highly branched neurites and a wide distribution in the tissue (Fig. 6k, m). In the same location, SC-NES cells formed small tight clusters with short linear projections and a drastically reduced spreading in the parenchyma (Fig. 6l–m).

Our data show that the integration of human NES cells into a SCI model is strictly related to cellular regional identity and that the anatomical match of neural implants with the recipient tissue is a fundamental requirement for connectivity.

**Global transcriptome analysis of grafted human SC-NES cells**. To analyze molecular differences between SC- and NCX-NES cells associated with their radically different adaptation in the lesioned SC, we performed a global RNA-sequencing (RNA-seq) analysis of tissue from both SC- and NCX-NES grafts. In order to segregate intrinsic from extrinsic factors contributing to graft integration, differentially expressed (DEX) genes were also compared to the transcriptional profile of SC- and NCX-NES cells in vitro before and after neural differentiation (Fig. 7a). Principal component analysis (PCA) of SC- and NCX-NES cells confirmed a clear separation of groups (Fig. 7b). In particular, we identified a first principal component (PC1) segregating in vitro from in vivo samples, whereas a third principal component (PC3) separates samples based on anatomical derivation (Supplementary Fig. 6a-b). Gene-ontology analysis showed that DEX genes upregulated in SC-NES cell grafts belonged to the "neurogenesis" ontology

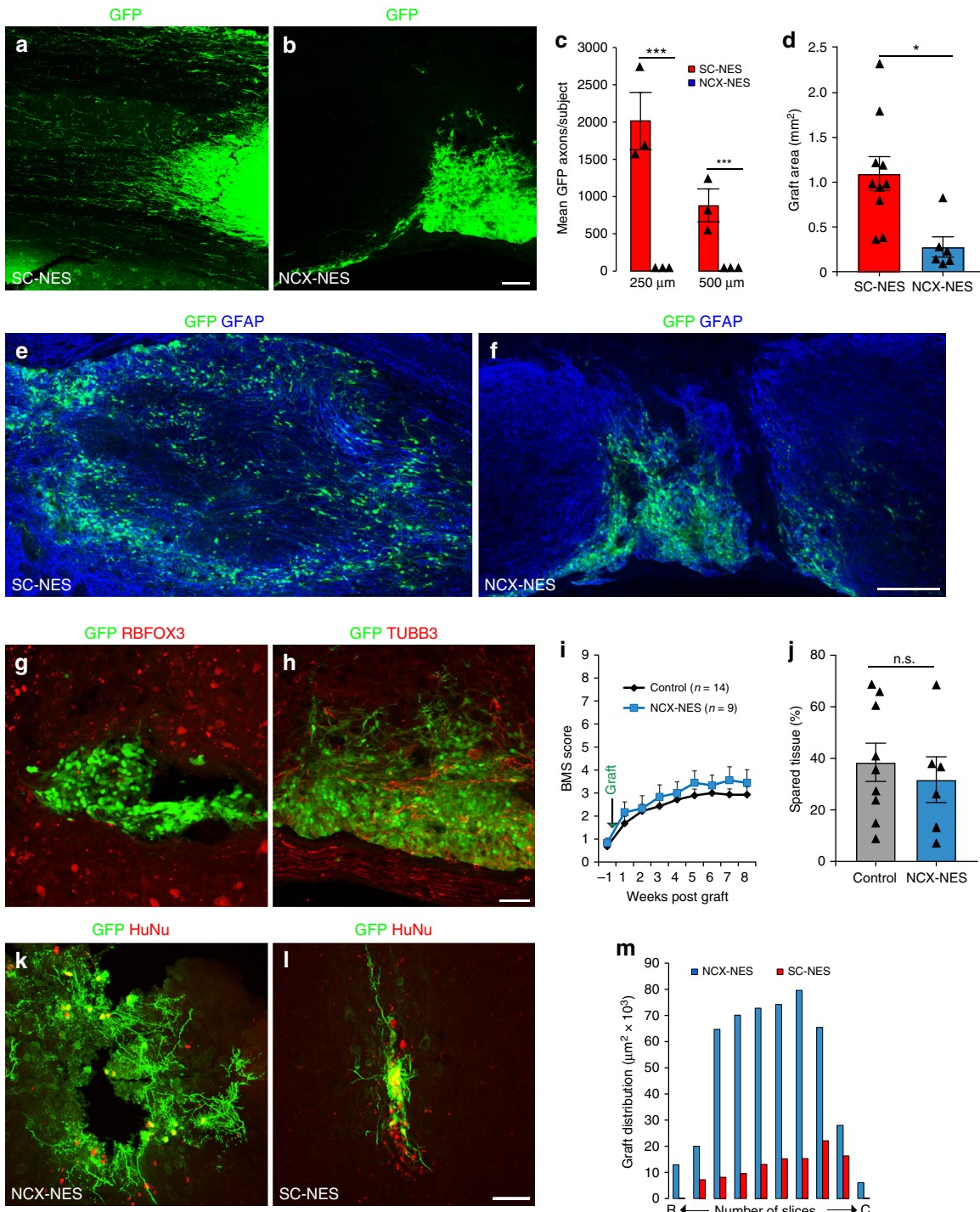

**Fig. 6** Regionally matched NES cell grafts integrate in the lesioned SC. **a** GFP immunostaining of human SC-NES cells grafts in the lesioned SC 8 weeks after implantation. SC-NES cells integrate in the tissue and show a massive axonal elongation. **b** GFP immunostaining of human NCX-NES cell grafts. The cells form tight clusters in the host SC with a poor axonal extension. **c** Quantification of GFP-positive axons per subject in SC- and NCX-NES cell transplants. The plot indicates the number of axons at 250 and 500 μm from the caudal graft-host interface. **d** Quantification of SC- and NCX-NES cell graft areas. **e, f** GFP and GFAP staining of SC- and NCX-NES cell grafts at the injection site showing a larger extension of SC-NES cell grafts in the host tissue. **g, h** Grafted NCX-NES cells are negative for the pan-neuronal markers RBFOX3 and TUBB3. **i** Hindlimb locomotion: BMS scores of NCX-NES cell recipient animals ($n = 9$) and control animals ($n = 14$) suggesting a lack of functional recovery in treated animals. **j** Spared tissue quantification in control and NCX-NES cell grafted mice. The histogram shows that the extent of the lesion in both groups is not statistically different. **k, l** GFP and human nuclei (HuNu) immunostaining of NCX-NES cells implanted in the motor cortex of immunodeficient mice two months after surgery. **m** Rostro (R)-caudal (C) spread and graft extension in the coronal plane two months after engraftment of SC- and NCX-NES cells into the motor cortex of adult immunodeficient mice. Data are expressed as mean ± s.e.m (*$P < 0.05$; ***$P < 0.001$; Student's $t$-test). Scale bars: **a, b**, 500 μm; **e, f**, 200 μm; **g, h, k, l**, 50 μm

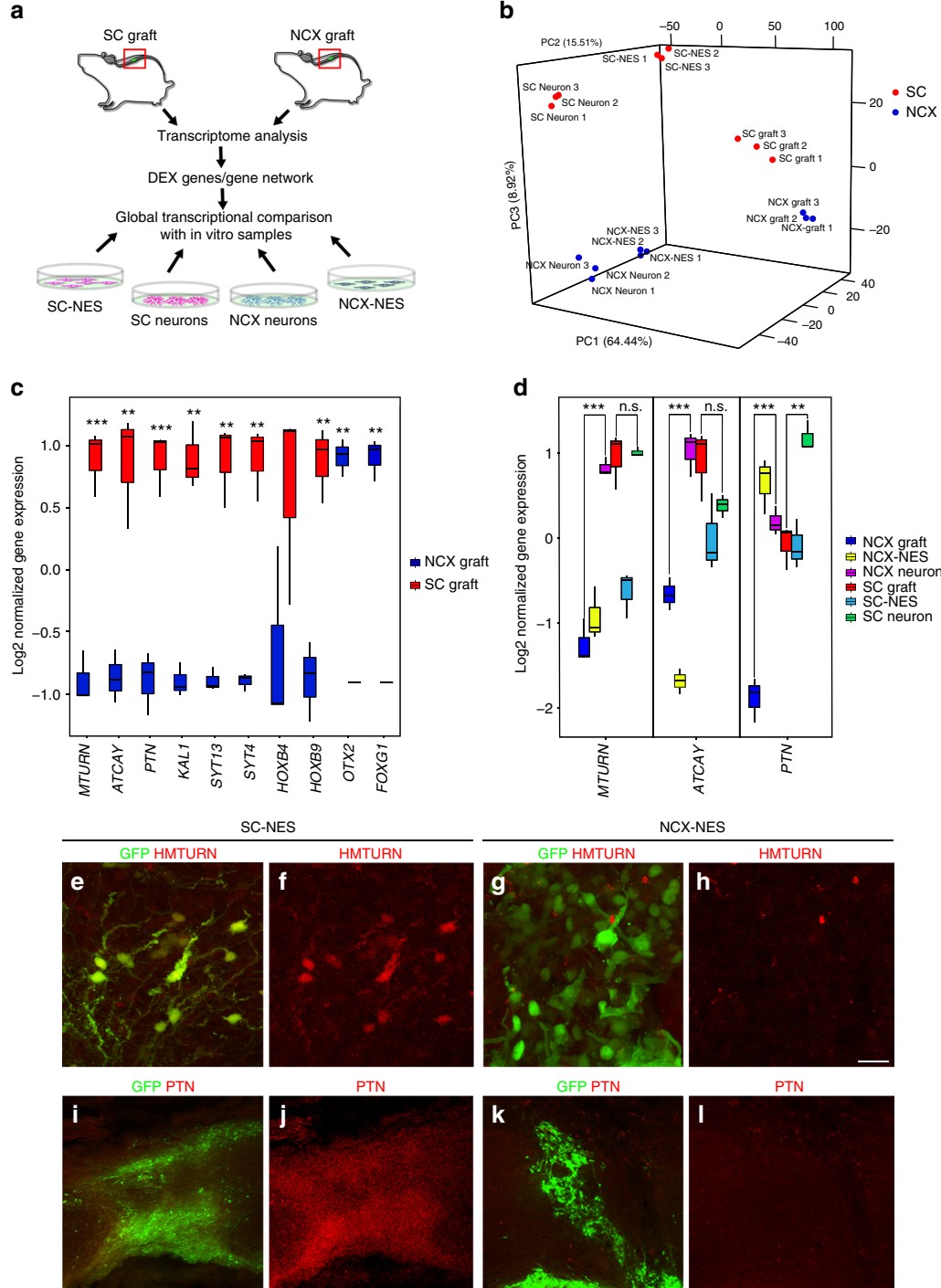

**Fig. 7** Global transcriptome analysis reveals SC-NES cell neuronal commitment after transplantation. **a** Schematics of the experimental procedure. SC- and NCX-NES cell grafts were isolated from recipient mice 8 weeks after implantation. RNA-seq was performed to identify differentially expressed (DEX) genes in the two samples. The expression level of DEX genes was eventually compared with in vitro samples: proliferating cells from both SC and NCX (SC-NES; NCX-NES), and cells upon 8 weeks of differentiation (SC Neurons; NCX Neurons). **b** 3D plot of the principal component analysis (PCA) of SC- and NCX-NES cells in proliferation (NES), after in vitro differentiation (neuron) or in vivo transplant for 8 weeks (graft). Each dot represents a sample. **c** Box plot of normalized gene expression for selected genes in SC- and NCX-NES cell grafts. The black bar within each box represents the median value. Vertical bars indicate the maximum and minimum values. **d** Box plot of normalized gene expression of maturin (*MTURN*), caytaxin (*ATCAY*) and pleiotrophin (*PTN*) genes across all SC ad NCX samples. The black bar within each box represents the median value. Vertical bars indicate the maximum and minimum value. **e–h** Human MTURN (HMTURN) and GFP stainings of SC- and NCX-NES cells 8 weeks after engraftment. Samples derived from the SC express HMTURN after grafting, whereas grafted NCX cells are HMTURN-negative. **i–l** Human PTN (PTN) and GFP stainings confirming the expression of PTN in grafted SC-NES cells and its absence in grafted NCX-NES cells. Scale bars: **e–h** 20 μm; **i–l** 50 μm. (\*\*$P < 0.01$, \*\*\*$P < 0.001$; Student's *t*-test)

cluster (GO: 0007399; $P$ value $1.3 \times 10^{-2}$), supporting our histological data reporting the in vivo neural differentiation of SC-NES cells compared to undifferentiated NCX-NES cells. Among the top 50 up-regulated genes in the SC-NES cell grafts were genes related to neuronal maturation, such as maturin (*MTURN*)[30] and caytaxin (*ATCAY*)[31], as well as genes involved in axonal elongation (pleiotrophin, *PTN*)[32] and cell adhesion (anosmin, *KAL1*)[33]. The maturation of SC-NES cells was also confirmed by the up-regulation of synaptic genes *SYT4* and *SYT13*, reinforcing our observation of graft–host interaction (Fig. 7c). Differential expression of SC- and NCX-positional genes, such as *HOXB4* and *HOXB9* or *FOXG1* and *OTX2* respectively, confirmed the long-term preservation of positional identity after transplantation (Fig. 7c).

Interestingly, while *MTURN* and *ATCAY* were upregulated by in vitro NCX-NES cell-derived neurons, their expression was strongly reduced in the grafts. On the other hand, the expression of these genes was maintained by SC-NES cells during engraftment, with no significant difference between SC-NES cell-derived neurons and SC-NES cell grafts (Fig. 7d). *PTN* expression was drastically reduced in NCX-NES cell grafts compared to NCX-NES cell-derived neurons, with less reduction observed in SC-NES cell grafts compared to the same cells after in vitro neural differentiation (Fig. 7d). Finally, we validated our transcriptome analysis with immunostainings for MTURN and PTN. The cytosolic protein MTURN was visualized in the soma and neurites of grafted SC-NES cells (Fig. 7e–f), whereas PTN was secreted into the extracellular environment (Fig. 7i–j). Neither protein was detected in the NCX-NES cell grafts (Fig. 7g–l).

To extract additional biologically relevant information, we applied weighted gene co-expression correlation network analysis (WGCNA) to identify gene modules with similar variation across in vitro and in vivo samples and identified overall 23 modules (Supplementary Fig. 7, Supplementary Data 1). We observed that module 1 (M1) identifies genes upregulated in grafted cells from both NCX and SC, suggesting that implantation itself impacts NES cell transcriptional profile. Genes in M9 were upregulated in grafted SC-NES cells and mostly related to synaptic signaling (GO: 0099536; $P$ value $4.48 \times 10^{-14}$), ion transmembrane transport (GO: 0034220; $P$ value $3.48 \times 10^{-10}$), neurogenesis (GO: 0048666 $P$ value $1.3 \times 10^{-6}$) and neuron projection development (GO: 0031175; $P$ value $4.43 \times 10^{-6}$). Module M8, on the other hand, clustered genes related to forebrain neuronal differentiation, revealing that while NCX-NES cells have the potential to differentiate, they fail to do so upon implantation into SC.

Thus, global RNA profiling provides strong evidence for selective integration into SC of human SC-NES cells as compared to NCX-NES cells. The findings indicate that upregulation of genes involved in neural precursor maturation, neurite extension, and synapse formation plays a crucial role in graft integration, and that host environment determines the transcriptional profile of implanted cells. Therefore, SC-NES cell integration results from both intrinsic and extrinsic factors, with an anatomical match of grafted cells to recipient tissue being required for successful SCI treatment.

**Transplantation in a model of spinal cord contusion**. To explore the translational impact of our findings, we challenged SC-derived NSCs with an allograft procedure in a clinically relevant model of SCI. For this purpose, we derived NSCs from the embryonic SC primordia of Sprague Dawley rats (r-NSCs) ubiquitously expressing the GFP reporter gene and transplanted into rats with a contusion of thoracic SC. We used rats as recipients since contusion is more controllable and reproducible in

larger species, and we administered cyclosporine A as immunosuppressant in place of genetic immunodeficiency. Anatomical analysis 2 months after engraftment confirmed that r-NSCs successfully survived in host SC, filled the contusion cavity and elongated bilaterally many GFP-positive axons, most of which were myelinated by host oligodendrocytes (Supplementary Fig. 8a-d). No graft rifts were observed. Functional analysis by Basso Beattie and Bresnahan (BBB) test[34] revealed improved motor performance in the transplanted group as compared to control rats, which reached significance at 7 and 8 weeks post-graft (Supplementary Fig. 8e). These results confirmed that SC-derived NSCs yield benefit in a clinically relevant SCI model.

## Discussion

In this study, we report the derivation and characterization of human SC-NES cells as candidates for cell-based therapy in SCI treatment. Previous investigations on SC-derived NSCs were limited to the application of one single fetal cell line (566RSC)[1,2,7] or to fetal NSCs cultured as neurospheres with the consequent reduction of the neurogenic potential[4,5]. Here we show that SC-NES cells can be successfully derived from human SC primordia and can be cultured long-term in adhesion without genetic immortalization. Further, we demonstrate that human SC-NES cells are highly expandable and retain over time intact karyotype and neurogenic capacity. In vitro experiments confirm that SC-NES cells are immunopositive for canonical NSC markers and can be efficiently differentiated toward the neuronal and glial fate. Importantly, SC-NES cells retain their regional commitment after propagation as demonstrated by the expression of specific transcription factors and the neurochemical identity of neural progeny[35,36].

When transplanted into injured SC, SC-NES cells exhibit excellent integration properties: they differentiate into neurons and extend long-distance axons reaching the rostral and caudal extremities of the recipient SC. Previous work has emphasized the importance of mechanical and trophic support to guarantee cell survival and neurite extension from grafted cells[1,11]. Human SC-NES cells, on the other hand, are intrinsically prone to elongate long distance axonal connections with no need of growth factor-enriched matrices. We also observed that SC-NES cells produce an amelioration of locomotor deficits and that the extent of the functional improvement depends on lesion severity. The infiltration of collagen and the subsequent formation of an impervious fissure within some grafts may have attenuated the behavioral outcome. However, the formation of a rift could be due to the hemisection surgical procedure which inevitably compromises meningeal integrity. In fact, no rifts were observed in the SC contusion experiment where *dura mater* was left intact. However, since contusion is the most common type of clinical SCI lesion, the formation of the rift may not be a limiting factor for potential clinical applications.

The relevance of graft rostro-caudal origin has not been systematically explored with human cell sources, though brain precursor transplantation has been described[37,38]. For rodent neural precursor transplantation, there has been a direct comparison of cells established from SC versus hindbrain versus telencephalon[6] and substantially greater host CST growth was observed with SC-matched transplants. However, the relative degree of engraftment for different source cells, and the converse engraftment of different source cells into NCX have not been reported previously. Here, in order to assess whether the engraftment potential of SC-NES cells is correlated to their anatomical derivation, we challenged NCX-NES cells with an identical experimental paradigm. Interestingly, we found that NCX-NES cells aggregate, forming dense clusters in the SC with a remarkably poor graft neurite

extension. On the other hand, when placed into NCX, they distribute widely and elongate branched projections suggestive of efficient integration. Thus, matching NES cell sources to engraftment site is essential not only for host CST axon growth into the graft, but also for human neuron engraftment per se, and most critically for graft axon extension.

To investigate the molecular mechanism responsible for the striking difference in the engraftment properties of SC- and NCX-NES cells, we performed global transcriptional investigation of implanted cells. Grafted SC-NES cells upregulate genes related to neuronal maturation (*MTURN, ATCAY*), cell adhesion (*KAL1*), axonal elongation (*PTN*), and synapse establishment (*SYT13, SYT4*). These six genes could be considered as positive markers of engraftment in order to delineate the molecular signature of cell populations suitable to achieve a successful integration. Moreover, release of secreted PTN into cerebrospinal fluid may provide an engraftment biomarker monitored by lumbar puncture during future transplantation studies. Two negative markers of engraftment have also been proposed by Ladewig and colleagues[39] who found that the secretion of FGF2 and VEGF by implanted neural progenitors impairs the integration in the tissue through chemoattraction. Additional studies may determine whether these markers could be used to screen cell populations before their implantation, thereby optimizing the experimental procedure and facilitating clinical transition. Such biomarkers may be especially relevant in the development of novel differentiation protocols from patient-derived pluripotent stem cells in order to enhance their integration potential and maximize their potential benefit.

The beneficial effect of NSCs in rewiring the lesioned SC occurs through a range of mechanisms including growth factor secretion[40–42], severed axon myelination[43], and enhancement of severed axon regeneration[1,6,11]. Grafted cells can provide a scaffold for severed axons to regenerate, and descending fibers from the CST, rubrospinal or raphespinal tracts have been observed to enter the graft[1,6,11]. However, the extent of axonal regeneration beyond the lesion is much reduced if not completely absent, and it is unlikely to support a functional benefit. It has also been proposed that NSCs can form relay circuits in the injured SC which are comparable to those formed by propriospinal circuits after incomplete SCI[44–47]. One key aspect of a relay system is that injured axons do not need to regenerate long distances and the onus for axon growth is placed on the transplant-derived neurons, which can be selected for their ability to extend long axons in vivo. Nevertheless, no proof has been provided previously to demonstrate a graft-mediated neural relay function. Using a pharmaco-genetic approach to obtain a transient silencing of implanted cells, we provide the first evidence that human SC-NES cells establish a functional circuit in recipient SC. Interruption of the circuit by the temporary silencing of SC-NES cell-derived neurons resulted in loss of the functional benefit in three different behavioral paradigms. Thus, SC-NES cells restore the disrupted connectivity by forming an intermediate station between the rostral and caudal segments of damaged SC.

The transition of human cell therapy to the clinic will likely require more systematic investigations. While recent studies show that human cell transplantation is possible in non-human primates[48], practical issues remain to be addressed. For instance, further studies are needed to understand guidance of extending axons to appropriate targets and to determine what effect newly generated circuits may have for autonomic and sensory function.

Our investigation of human NES cells provides tools to understand the molecular features necessary to achieve successful engraftment in terms of both efficient cell integration and host axonal growth. Having the means in hand to delineate the transcriptional profile of human cells with a high engraftment potential provides an important instrument for future development in the field.

## Methods

**Human tissue procurement**. All work was performed according to the NIH guidelines for the acquisition and distribution of human tissue for bio-medical research purposes and with the approval by the Human Investigation Committees and Institutional Ethics Committees of each institute from which samples were obtained. De-identified postmortem human brain and SC specimens were provided by the Joint MRC/Wellcome Trust (grant #099175/Z/12/Z) Human Developmental Biology Resource (http://hdbr.org). Appropriate informed consent was obtained and all available non-identifying information was recorded for each specimen. Tissue was handled in accordance with ethical guidelines and regulations for the research use of human brain tissue set forth by the NIH (http://bioethics.od.nih.gov/humantissue.html) and the WMA Declaration of Helsinki (http://www.wma.net/en/30pubblications/10policies/b3/index.html).

**Human NES cell derivation**. De-identified prenatal human samples were staged, as previously described[49]. Human NES cells were derived from dorsal forebrain and SC of six post-mortem specimens ranging from 5 to 8 PCW (Carnegie stages 15–23). After removal of all meninges, the NCX and SC tissues were dissected into small fragments (~1 mm²) in cold PBS. Then, samples were collected by centrifugation at $150 \times g$ for 3 min. The pellets were incubated on ice for 15 min, gently flicking every 2 min. Subsequently, samples were incubated with 0.05% trypsin (Gibco, #25200056) at room temperature for 1–2 min and dissociated by pipetting up and down every 30 s until a single cell suspension was obtained. Trypsin was inactivated with PBS supplemented with 10% fetal bovine Serum (FBS, Invitrogen, #16140-071) and cell suspension was centrifuged at $150 \times g$ for 3 min. The cell pellet was resuspended with NES medium prepared with DMEM/F12 (Gibco, #PHG0311) with the addition of B27 supplement (1:1000, Invitrogen, #175040-44), N2 supplement (1:100 Gibco, #17502-048), 20 ng/ml FGF2 (Gibco, #13256-029), 20 ng/ml EGF (Gibco, #PHG0311), 1,6 g/l glucose, 20 µg/ml insulin (Sigma, #I9278) and 5 ng/ml BDNF (R&D Systems Inc, #248-BD-01M). 10 µM Rock Inhibitor (Y-27632, Stemgent, #04-0012) was added to increase cell viability. The cells were then plated onto dishes coated with poly-L-ornithine (0.01%, Sigma, #P4957), laminin (5 µg/ml, Invitrogen, #23017-015) and fibronectin (1 µg/ml, Corning, #354008). Coated dishes were incubated for 1 h at 37 °C and rinsed three times with distilled water prior to be used for cell culture. Medium was changed 12 h after plating and the cells were passaged using 0.25% trypsin within 3 days and replated to obtain a monolayer of cells at a density of ~10⁵ cells/cm². Half of the media was changed every 2–3 days to allow culture conditioning. In the transplantation studies, we used one male SC-NES cell line (HSB318) and one female NCX-NES cell line (HSB325).

**Maintenance and differentiation of NES cell lines**. NES cells were kept in proliferation in T25 flasks coated with poly-L-ornithine (0.01%), laminin (5 µg/ml) and fibronectin (1 µg/ml). Cells were expanded in NES medium and reached confluency ($0.5–1 \times 10^5$ cells/cm²) in about 7 days. Cells were split 1:2 every 5–7 days with 0.25% trypsin. The expansion rate of the cells was constant over time. Half volume of the medium was changed every 2–3 days. For cryopreservation, cells were trypsinized and ~$2 \times 10^6$ cells were pelleted ($200 \times g$ for 3 min) and resuspended in cryopreservation medium containing 10% DMSO and 90% NES medium. For resuscitation, cells were rapidly thawed at 37 °C, resuspended in pre-warmed culture media, centrifuged for 3 min at $200 \times g$, resuspended in NES medium, and replated on coated plates. Neuronal differentiation of NES cells was performed in two steps. For the pre-differentiation step NES cells were seeded at a density of $0.5 \times 10^5$ cells/cm² in a T25 coated flask in NES medium without EGF and FGF2. After 7 days, cells were dissociated and replated at a density of $0.8–1 \times 10^5$ cells/cm² in BrainPhys Medium (Stem Cell Technologies, Cambridge, MA, #05790) supplemented with B27 (2%), N2 (1%) and BDNF (20 ng/ml). Half volume of the medium was changed every 2–3 days and neurons were differentiated up to 3 months.

**Animals**. Mice: 8–10 weeks old female NOD.Cg-*Prkdc*^scid^ *Il2rg*^tm1Wjl^/SzJ mice (Jackson Laboratories) were used in the present study as NES cell recipients. *Rats*: SD-Tg(CAG-EGFP)CZ-004 Osb rats were a generous gift from Dr. Jeffery Kocsis (Yale University, Department of Neurology) and were used to derive GFP-positive NSCs. At 10–11 weeks old female wild type Sprague Dawley rats were used as recipients for rat NSCs grafts. All animals were housed on a 12 h light/12 h dark cycle and had free access to food and water throughout the study. All experimental procedures were performed in compliance with animal protocols approved by the Institutional Animal Care and Use Committee at Yale University.

**Karyotype analysis**. Karyotype analysis was performed by core facility at the Department of Genetics at the Yale School of Medicine. Briefly, the cells were cultured in the presence of 0.1 µg/ml colcemid for up to 4 h and then fixed. Metaphase spreads were analyzed.

**Electrophysiology**. Electrophysiological experiments were performed on differentiated SC-NES cells infected with the hM4Di-mcherry lentiviral construct. Recordings were performed at 21–23 °C using the whole-cell patch-clamp technique in voltage- and current-clamp configurations. Cells were continuously perfused with an extracellular solution composed by 125 mM NaCl, 2.5 mM KCl, 26 mM NaHCO$_3$, 15 mM glucose, 1.3 mM MgCl$_2$, 2.3 mM CaCl$_2$, 1.25 mM NaH$_2$PO$_4$ (bubbled with 95% O$_2$, 5% CO$_2$) and were visualized using an Axioskop FS2 microscope (Zeiss) equipped with epifluorescence and infrared differential interface contrast (DIC) optic. An X-Cite 120LED lamp (Excelitas) and an appropriate filter set were used to identify mcherry-positive cells. Pipettes were produced from borosilicate glass capillary tubes (Sutter Instruments) by mean of a horizontal puller (P-2000, Sutter instruments) and filled with the following intracellular solution: 130 mM K-gluconate, 4 mM NaCl, 2 mM MgCl2, 1 mM EGTA, 5 mM creatine phosphate, 2 mM Na2ATP, 0.3 mM Na3GTP, 10 mM HEPES (pH 7.3 with KOH). Membrane voltage was corrected off-line for a calculated liquid junction potential of −10 mV. Series resistance was always compensated by 70–80% and monitored throughout the experiment. Clozapine-N-oxide [10 μM] (Sigma, St. Louis, MO #C0832) was applied in the bath through perfusion. Recordings were made with a MultiClamp 700B amplifier (Molecular Devices) and digitized with a Digidata 1322 computer interface (Molecular Devices). Data were acquired at a sampling frequency of 20 kHz and filtered at 10 kHz using the software Clampex 9.2 (Molecular Device). Software Clampfit 10.2 (Molecular Devices) and OriginPro 8 (Microcal) were used for data analysis.

**Immunofluorescence**. NES cells were washed with PBS and fixed with 4% formaldehyde for 10 min at RT. After three additional washes in PBS, cells were left in blocking solution (PBS supplemented with 1% horse serum and 0.1% Triton) for 1 h at RT. Cells were incubated with primary antibodies o/n at 4 °C. Primary antibodies were diluted as follows: Nestin (1:200 R&D systems #MAB1259); PAX6 (1:200 BD Bioscience #561462); phospho-vimentin (1:200 Abcam #ab22651); SOX2 (1:400 Millipore #ab5603); vimentin (1:200 Abcam #ab16707); CHAT (1:500 Millipore #AB144P); Doublecortin (1:200 Cell Signaling #4604); GFAP (1:500 Sigma #G3893); RBFOX3 (1:500 Millipore #MAB377); TUBB3 (1:1000 Abcam #ab107216); Neurofilament (1:200 Millipore #MAB1615); MAP2 (1:1000 Millipore #AB5622); O4 (1:200 R&D systems #MAB1326) PSD95 (1:250 Thermo Fisher Scientific #51-6900); Synaptophysin (1:500 Millipore #MAB329); KI67 (1:500 Abcam #ab16667); HOXB9 (1:200 Abcam #ab66765). The following day, cells were washed in PBS three times before being incubated with Alexa 568 or Alexa 488 secondary antibodies (1:500 Invitrogen) for 1 h at RT. The samples were finally mounted using mounting medium (Life Technologies #P36935) containing DAPI.

**Western blot**. NES or HEK 293 cells were washed with PBS and harvested in radioimmunoprecipitation assay (RIPA) buffer. WT or NgR1 KO mouse forebrain were dissected and homogenized in RIPA buffer. The samples were then centrifuged at 20,000 × g for 20 min. The lysate was resolved by SDS-PAGE, transferred to nitrocellulose membranes, immunoblotted with anti-β-actin (1:3000 Sigma-Aldrich, #A1978), and anti-NgR1 (1:1000 R&D Systems, AF1440) primary antibodies o/n at 4 °C. After primary antibody incubation, secondary antibodies (Odyssey IRDye 680 or 800) were applied for 1 h at room temperature. Membranes were then washed and visualized using a Licor Odyssey Infrared imaging system.

**Thoracic dorsal hemisection surgery**. Mice were first anesthetized with 4% isofluorane and maintained with 2% isofluorane throughout the whole procedure. A laminectomy was performed to expose the dorsal portion of the SC corresponding to the T8 and T9 levels. The dorsal hemisection was performed at the T8 level with a pair of microscissors to a depth of 1.1 mm to completely sever the dorsal and dorsolateral CST. The lateral portions of the SC were scraped with a 30 gauge needle to ensure the completeness of the lesion. Muscle and skin overlying the lesion were sutured with 4.0 vicryl. All animals received subcutaneous injection of 100 mg kg$^{-1}$ ampicillin and 0.1 mg kg$^{-1}$ buprenorphine twice a day for the first two days after surgery. Bladders of injured animals were expressed manually on a daily basis throughout the whole experiment.

**SC contusion surgery**. Female Sprague–Dawley rats (10–11 weeks, 220–240 g) were used in this study. Animals were anesthetized with intraperitoneal injection of ketamine (60 mg kg$^{-1}$) and xylazine (10 mg kg$^{-1}$) mixture. A laminectomy was conducted at the caudal portion of T6 and all of T7 spinal levels. A T7 severe contusion injury (weight of 10 g, height of 50 mm) was produced with the MASCIS impactor. After the spinal contusion, muscle and skin layers were sutured with 4.0 polyglactin. Bladders of injured animals were expressed manually twice a day throughout the whole experiment.

**Cell engraftement**. Ten days after mouse dorsal hemisection surgery, lesioned animals underwent a second procedure for NES cell implantation. Mice were anesthetized with 4% isofluorane and maintained anesthetized with 2% isofluorane until the procedure was completed. The original incision was reopened and the SC re-exposed. NES cells were resuspended in BrainPhys medium supplemented with 10 μM Rock Inhibitor and 20 ng/ml BDNF at a density of 250,000 cells/μl. NES cells were kept on ice throughout the procedure. Two injections were performed on

the injury site, each delivering 1 μl of cell suspension at a distance of ~1 mm one from the other. The injection was performed using a glass capillary attached to a syringe (Hamilton, Reno, NV) and a micropump (Ultramicropump III, World Precision Instruments, Sarasota, FL). The rate on the injection was 250 nl/min and the capillary was left in place for two additional minutes before withdrawal. Control animals underwent the same surgery and received a vehicle injection. At the end of the procedure, the muscle and the skin were sutured with vicryl 4.0 and animals received ampicillin 100 mg kg$^{-1}$ subcutaneous injection for 3 days. Animals survived 8 weeks before being sacrificed for the histological analysis.

Nine days after rat SC contusion, animals underwent surgery for NSCs engraftment. Embryonic day 13.5 (E13.5) SC from transgenic SD-EGFP rats provided donor tissue for grafting. Rat NSCs were resuspended in a fibrin matrix enriched with growth factors, as previously described[1]. Starting 24 hours before cell transplantation and throughout the whole experiment rats were treated with cyclosporine A (15 mg kg$^{-1}$) as immunosuppressant drug administered i.p. on a daily basis. The graft mixture (250,000 cells/μl) was microinjected into three sites on the lesion delivering 3 μl per injection. Animal underwent functional testing for up to 8 weeks and were sacrificed for the anatomical analysis through transcardial perfusion with 4% formaldehyde.

For NCX transplantation, mice were anesthetized with 4% isofluorane as previously described. A craniotomy was performed using a micromotor system (Foredom, Bethel, CT) exposing the motor cortex. The cortex was damaged by aspiration of a portion 1 × 1 mm$^2$. A volume of 1 μl of NES cell suspension (250,000 cells/μl) was injected onto the damaged area using a glass capillary attached to syringe (Hamilton, Reno, NV) and a micropump (World Precision Instruments, Sarasota, FL). Animals were kept for 8 weeks before being sacrificed for histological analysis.

**Anterograde labeling of the CST**. Six weeks after injury, descending CST fibers of grafted mice were labeled with BDA (0.1 g/ml in sterile normal saline, Thermo-Fisher Scientific # D1956) by injection into five spots of the right motor cortex. The skulls of anesthetized mice were tightly fixed to a stereotaxic apparatus (Kopf, Tujunga, CA). Craniotomy over the motor cortex area was carried out using a micromotor system (Foredom, Bethel, CT). The injection area on the right hemisphere was defined in a rectangle with one side measuring 2 mm (from 1.0 mm anterior to −1.0 mm posterior to the bregma) and one side measuring 1.5 mm (lateral to the bregma). Injections were performed using a glass capillary attached to a microsyringe (Hamilton, Reno, NV) at a 0.7 mm depth. Each injection delivered 75 nl of BDA solution into the motor cortex at a rate of 75 nl/min. The tip of the glass capillary was left in place for two additional minutes before withdrawal. Two weeks later, the animals were sacrificed by transcardial perfusion with PBS followed by 4% formaldehyde. To visualize the BDA, a tyramide signal amplification fluorescence system (Perkin Elmer, Waltham, MA) was used.

**Immunohistochemistry**. After perfusion tissues were removed, post-fixed overnight in 4% formaldehyde at 4 °C, and cryoprotected in 30% sucrose (Sigma-Aldrich, St. Louis, MO) until they sank before being sectioned on a Leica cryostat. SC sections were cut horizontally at a thickness of 30 μm. Brain sections were cut sagitally at a thickness of 40 μm. Sections were washed three times in PBS between each step in the immunodetection protocol. For detection of the antigens with fluorescent antibodies, the protocol was performed as follows: sections were left in blocking solution (PBS supplemented with 1% horse serum and 0.1% Triton) for 1 h at RT and then incubated overnight at 4 °C with primary antibodies diluted in blocking solution. Primary antibodies were diluted as follows: APC (1: 200 Millipore #OP80); active (cleaved) Caspase 3 (1:150 Millipore #AB3623); CHAT (1:500 Millipore #AB144P); Collagen type IV (1:500 Biogenex #AM379-10M); Doublecortin (1:200 Cell Signaling #4604); GAD1 (1:250 R&D systems #AF2086); GFP (1:1000 Life Technologies #A11122 or SantaCruz Biotechnologies #sc-9996); GFAP (1:500 Sigma #G3893); HOXB9 (1:200 Abcam #ab66765); Human GFAP (1:1000 Takara #Y40420); Human Neurofilament (gift from Dr. Virginia Lee and Dr. John Trojanowski, University of Pennsylvania, used undiluted); Human Nuclei (1:200 Millipore #MAB1881); Human NCAM (1:1000 SantaCruz Biotechnologies #sc-106 incubated at RT); Human Synaptophysin (1:1000 eBioscience #14-6525-82); KI67 (1:500 Abcam #ab16667); Nestin (1:200 R&D systems #MAB1259); Human MTURN (1:1000 Thermo Fisher Scientific # PA5-56177); RBFOX3 (1:500 Millipore #MAB377 or #ABN78); MAG (1:200 Millipore #MAB1567); Pleiotropin (1:100 abcam #ab93685); SOX10 (1:250 Abcam #ab155279); Synaptophysin (1:1000 Millipore #MAB329); TUBB3 (1:1000 Abcam #ab107216); Vglut2 (1:1000 Millipore #MAB5504); 5HT (1:10000 Immunostar #20080); Alexa 568-conjugated streptavidin (1:500 Thermo Fisher Scientific #S11226) to label BDA traced CST axons. The following day, sections were incubated for 1 h at RT with appropriate Alexa 488, Alexa 568 or Alexa 647 (1:500 Invitrogen) secondary antibodies. The sections were finally covered with mounting medium (Life Technologies #P36935) containing DAPI before being coverslipped. All images were acquired using a Carl Zeiss LSM 710 confocal microscope. Image processing was performed using Zeiss Zen, Adobe Photoshop and Illustrator.

**Quantification of cell types in the graft and axons**. Cellular differentiation in the grafts was determined by counting individual cells labeled for RBFOX3, GFAP,

KI67, CHAT or caspase 3 in 8 randomly selected fields (4 sections per animal, 2 fields per section) and expressing them as a percentage of the total number of human nuclei positive cells in the field. Cells were visualized using a Carl Zeiss LSM 710 confocal microscope at a magnification of ×63. The counting was performed using the built-in plugin of the Image J program. The number of GFP labeled human axons emerging from the graft was quantified using a Carl Zeiss LSM 710 confocal microscope. For every 6th consecutive horizontal section, a mediolateral line was drawn 250 and 500 μm caudal to the graft/host interface under a ×40 magnification. The tissue was then examined under a ×600 magnification and GFP labeled axons that intersected this line were marked and counted. To estimate the total number of axons/subjects, the number of axons counted on sections was multiplied by 6. To visualize axons in the gray matter sections were co-stained with RBFOX3 antibody.

**Behavioral analysis**. A total of 64 mice underwent SC mid-thoracic dorsal hemisection studies in two batches. Forty-nine (49) mice underwent hemisection injury and were tested for their locomotor ability using the Basso Mouse Scale (BMS)[26] 7 days after lesion. Seven out of 49 mice were eliminated from further study because of unsuccessful hemisection surgery, defined by BMS score higher than 2 on day 7. The remaining 42 mice were randomized to receive vehicle injection (14 mice), SC-NES cell injection (19 mice) or NCX-NES cell injection (9 mice) at 10 days after hemisection. These 42 animals were followed by BMS behavioral test performed by two investigators blinded to the treatment for a total of 8 weeks after cell implantation, then received BDA injection and were processed for histological analysis. A post-hoc subgroup analysis of behavior in 25 (9 controls, 10 SC-NES and 6 NCX-NES cell recipients) out of 42 animals were used for spared tissue assessment and correlation studies.

In a separate experiment, 15 mice underwent hemisection injury and received SC injection of either hM4Di-expressing SC-NES cells (8 mice) or vehicle injections (7 mice) 10 days after hemisection. Their behavior was assessed by BMS, grid walking and footprint test with and without CNO administration 8 weeks after transplantation. Grid walking and footprint analysis were performed for animals able to walk, i.e. scoring at least 4 at the BMS test. For the footprint analysis animals had water-based nontoxic paint placed on the plantar surfaces of all four limbs (red on forelimbs and blue on hindlimbs) and were then run three times along a 1 m long narrow corridor lined with absorbent paper. Stereotyped gait and motor coordination parameters, including hindlimb stride length and hindlimb base of support, were measured from three complete step cycles from the middle of the runway. Animals were trained for one week in order to collect the baseline scores. To validate the behavioral outcome upon silencing NES cells, animals were given an i.p. injection of CNO, (1 mg kg$^{-1}$[50]) and were tested 30 min later. The wash-out scores were collected 24 h later. For the grid-walking test mice were placed on an elevated 45 × 45 cm metal grid with 2.5 × 2.5 cm square spacing covered by a dark box to make the environment more comfortable for the animals. Mice were videotaped via reflection from a mirror placed under the grid and allowed to explore the grid for 3 min. Videos were scored for the percentage of impaired steps out of the first 50 steps for each hindlimb individually. Impaired steps included a foot slip where the limb fell between the rungs or an incorrectly placed step where either the ankle or the tips of the digits were placed on the rung instead of proper grasping of the rung. Similar to the footprint test, baselines scores were collected after one week of training. The CNO effect was recorded 30 min after i.p. administration, whereas the drug wash-out assessment was performed on the following day.

Rats were evaluated using the BBB locomotor scale[34] two days before cell implantation and every week until killing.

**Transcriptome analysis**. For the transcriptome analysis each sample was handled in triplicate. Total RNA was extracted using the RNeasy plus minikit (Quiagen, #74134) according to manufacturer's instruction. mRNA libraries were prepared according to the TruSeq RNA Illumina kit protocol and sequenced using a HighSeq 2500 sequencing system (Illumina). The sequenced reads were clipped one nucleotide in both ends, leaving 74 nucleotide long reads for sequence alignment to human reference genome (GRCh38/hg38) by using the STAR software[51]. Beyond default parameters in the alignment procedure, human gene annotation retrieved from the GENCODE project (version 21, http://www.gencodegenes.org/releases/21.html) was additionally provided to improve exon–exon junction mapping. To quantify accurately the origin of the genomic locus of short sequenced reads, only the uniquely mapped reads were used for downstream analyses. *FeatureCounts* was used to quantify expression profiles of each type of annotation entry retrieved from GENCODE v21[52]. Additional quality control measures were introduced to assess potential issues including reads classification, mitochondrial contamination, and gene coverage uniformity. R package DESeq2 was used to identify DEX genes[49]. We generated two types of gene expression value, read counts and RPKM (reads per kilobase of exon model per million mapped reads). The reads count per gene served as the input for DESeq2. When performing the comparisons, DESeq2 first gets the mean expression level as a joint estimate for both groups, and then calculates the difference as well as the *P* value for the statistical significance of this change. The adjusted *P* value was calculated based on multiple testing with the Benjamini–Hochberg procedure, estimating the false discovery rate (FDR). To detect statistically significant DEX genes, FDR was set to be minor than 0.01. The

gene ontology (GO) analysis was performed using The Database for Annotation, Visualization and Integrated Discovery (DAVID) version 6.8.

**Weighted gene correlation network analysis (WGCNA)**. The R WGCNA package was used to find modules of highly correlated genes and characterize their expression patterns[53]. Genes expressed in at least 25% of samples were included for the analysis. Prior to module detection, scale-free topology was calculated to determine soft power, which was set at 6 and used to generate dissimilarity matrix. Then, module clustering was process based on dissimilarity matrix with minimum module size setting at 100. Detected modules were further merged together if they have very low dissimilarity (threshold = 0.2) and 23 modules were finally determined. To characterize the module expression pattern, we calculated the eigengene of each module via moduleEigengenes function. Eigengene expression pattern was shown using box plot.

**Statistical analysis**. For comparison between two groups two-tailed Student's *t*-test was used at a designated significance level of $P < 0.05$. Measurements taken at different time points were compared using the repeated measures one-way ANOVA. Analyses were conducted using Excel and SPSS. For the electrophysiology experiments, statistical significance was determined by one-sample Student's *t*-test. All data are presented as mean ± s.e.m. The statistical details of the experiments can also be found in the figure legends. No statistical methods were used to calculate sample size estimates. No animal exclusions were made except for the subgroup specified in Fig. 5d. Animals were allocated to injection groups in alternating order by an observer unaware of treatment. All behavioral and histological analyses were conducted without knowledge of treatment group.

**Data availability**. All relevant data that support our experimental findings are available from the authors. Supporting RNA-sequencing data are available on NCBI GEO (Gene Expression Omnibus), accession number GSE107514.

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

## Acknowledgements

We thank Dr. Trojanowski and Dr. Lee for providing the anti-human neurofilament antibody and Dr. Broccoli for the hM4Di-mcherry construct. We thank Dr. Lisgo for human tissue procurement. We also thank Hiam Naiditch and Tomoko Sekine-Konno for their contribution to the behavioral analysis. This material is based upon work supported by the State of Connecticut under the Regenerative Medicine Research Fund to M.T.D.'A and M.O. Its contents are solely responsibility of the authors and do not necessarily represent the official views of the State of Connecticut or Connecticut Innovations, Incorporated. This work was also supported by grants from the Wings for Life to X.W., from the Falk Medical Research Trust to S.M.S., and from the N.I.H. to S.M.S.

## Author contributions

M.T.D.'A. conceived and carried out the experiments, interpreted the results and wrote the manuscript. X.W. performed the spinal cord lesions. M.O. provided the NCX-NES cells and contributed to the interpretation of the transcriptome analysis. M.L. performed the RNA-seq data analysis, S.M. performed the WGCNA analysis, F.T. performed the electrophysiology experiments. Y.S and W.B.J.C. contributed to the behavioral analysis. F.L. assisted in cell culture work. N.S. and W.B.J.C. contributed to the manuscript preparation. S.M.S. contributed to the conception of the project and the interpretation of the results and wrote the manuscript.
