## [Peer Review File · Nature Communications]

Reviewers' comments:

Reviewer #1 (Remarks to the Author):

This manuscript reports on grafting human neuroepithelial stem cells (NEC) derived from either fetal spinal cord (SC-NEC) or fetal cerebral neocortex (NCX-NEC) and propagated and expanded in vitro and then grafted into spinal cord injury (SCI) lesions of immunodeficient (NOD/SCID) mice. The NEC were characterized extensively in vitro. The SCI lesions were dorsal hemisections, a form of incomplete SCI. The NEC suspended in medium plus BDNF were grafted into lesion sites 10 days after SCI.

The main findings reported are: (1) Human SC-NES are self-renewing in vitro and can be propagated and expanded in vitro for long periods. SC-NES can be differentiated to neurons and glia in vitro. (2) Human SC-NES survive well after grafting into immunodeficient mice and differentiate into propriospinal neurons that send axons into host tissue. (3) SC-NES derived neurons form contacts with host neurons. (4) SC-NES grafted mice exhibit some locomotor improvements after SCI. (5) NCX-NEC derived cells do not efficiently form neurons or improve function when grafted into spinal cord lesions, but do form neurons when grafted into cortex. (6) Transcriptome screening begins to identify molecular characteristics that may underlie specifications of neurons derived from NEC from different regions.

Technical aspects of the study appear well conducted and appropriately controlled. The data presented look to be of high quality. The microscopic images are consistent with the associated quantitative data. The data presented convincingly support the interpretations made in the text. Overall I found this to be a high quality study that provides useful and important new information about grafting human NEC in the context of SCI. In particular, the study provides clear evidence that regional specificity of NEC progenitor populations is critical in determining integration and function of graft-derived neurons, and begins to identify relevant molecular characteristics. Nevertheless, I have two important and not difficult to deal with points that the authors should attend to (and one minor point).

Specific comments:

1. Previous studies using fetal derived cells for grafting into spinal cord led to the problem of seeding distant ectopic colonies of grafted cells along the spinal canal (which provides a periependymal stem cell niche that can attract grafted cells). Some colonies were observed as far away as the 4th ventricle, and some along the surface of the spinal cord and brain. This observation led to some controversy and there have been a number of papers commenting on this. In the last few sentences of the discussion, the authors mention previous observations, and then briefly comment that they too observed such seeding of ectopic colonies of grafted cells, but provide no real detail. This topic deserves a separate (but short) section at the end of the results, including some data and possibly an image in a supplementary. As regards data, the authors should document what proportion of animals exhibited ectopic colonies (2 of 10 or whatever) and how many colonies were observed per animal and where those were located (2mm away from graft along central canal or cord surface or whatever), and how far away from the graft sites was routinely examined. This type of information is important for the field to progress and should not be glossed over.

2. A major part of the rationale for grafting neuronal progenitors from embryonic spinal cord rather than progenitors from some other CNS region after SCI derives from the notion that spinal cord progenitors give rise to propriospinal neurons, and that propriospinal neurons can form functionally meaningful relay circuits that can convey voluntary motor control across the injury site. To do so these neurons would need to receive inputs from descending projections from the brain and relay information conveying voluntary control to motor centers below the injury. The present authors seem to assume that this is a self-evident or long-standing concept, but it is not. As recently as 15 years ago, the notion intrinsic spinal cord neurons (propriospinal neurons) could create such a relay would have been scoffed at by motor physiologists, and consequently,

transplantation studies aimed at grafting neuronal progenitors were viewed with great skepticism at that time and were not vigorously studied. This view was changed by two landmark studies demonstrating that after SCI, descending brain projections form new connections with propriospinal neurons and that such connections can relay information. Bareyre et al (2004) showed that after partial SCI, descending projections from the brain can form new connections with propriospinal neurons and form new relay circuits that bypass the incomplete injuries. Courtine et al (2008) showed that after multiple staggered SCI lesions that completely remove all direct descending projections from the brain, the propriospinal network of neurons can form new relay circuits that are required and sufficient to convey voluntary locomotor commands from the brain to locomotor centers below the injuries. Without these studies, the type of grafting reported here would still be widely viewed with skepticism. Indeed, the terminology and concept of 'relay circuits' after SCI, which the current authors use in their title and discussions, was originated and validated by these studies. The present authors should cite these two studies and state that these studies underpin the rationale for transplanting NES that can form spinal cord intrinsic propriospinal neurons that have the potential to form functionally meaningful relay circuits after SCI. (The papers referred to in references 45 and 46 did not originate or validate this concept). This can only strengthen the notion that grafting regionally specific neuronal progenitors is important to achieve functional benefit as indicated by the authors results here.

Bareyre et al (2004) The injured spinal cord spontaneously forms a new intraspinal circuit in adult rats. *Nat Neurosci* 7:269-277.

Courtine et al (2008) Recovery of supraspinal control of stepping via indirect propriospinal relay connections after spinal cord injury. *Nature Medicine* 14:69-74.

3. Minor - The abstract should mention that the host animal for human NES transplantation in vivo was immunodeficient (NOD/SCID) mice (not just 'rodent').

Reviewer #2 (Remarks to the Author):

This is an interesting paper where the authors have isolated either spinal cord or cortical tissues from very young human embryos, expanded them in culture and then transplanted them in to a mouse model of SCI. The conclusions of this paper are that the expanded cells from the SC but not the Ctx can restore some function in this model. There is a correlation with specific gene sets. The implications are that they have a new potential cell therapy for SC in humans, with the caveat that some animals seemed to retain some rosettes and had some aberrant migration of the cells. Overall this was an interesting study from a very well-established investigator. However, there have been many similar studies previously with very similar results questioning the novelty of the findings (e.g., Kadoya et al, *Nat Med*, 2016). Furthermore, there are serious methodological and interpretational shortcomings that dampen enthusiasm for this report.

Major Concerns:

1. The authors need to show more details of their expansion of the NSC's to support the conclusion that they are stable (giving rise to similar numbers of neurons) and karyotypically normal at every passage. At minimum they should show expansion rates over time, sequential plate downs and counts of neurons/astrocytes produced, and at least three karyotypes at the start, middle (6 months) and end of their expansions. In addition the metaphase spread in Fig 1g looked squashed and it was difficult to read.
2. The authors did the best job they could with what was very weak functional data. This topic has been extensively discussed by NIH due to very variable results in the field - even to the point of generating testing centers for SC injury models and recovery (*Experimental Neurology*, Volume

233, Issue 2, February 2012, Pages 597- Replication and reproducibility in spinal cord injury research) The only significant effect came when they looked at the smallest lesions . This type of post hoc analysis while valid - speaks to the weakness of the data. It would be far more convincing to repeat the study focusing on producing milder lesions and establishing differences between transplanted and non transplanted groups. Post hoc decisions about cut off points (in this case 25%) can be a biased procedure based on particular experiments.

3 . The CNO experiment was a heroic attempt to show that connectivity was important. However, figures 5e and f were not convincing. There was certainly a trend down and up upon CNO administration suggesting an effect on function, but the control animals seemed to be extremely variable calling into question the reliability of the BMS score at 8 weeks . Combined with the weak overall effects of the transplant this further dampened enthusiasm. In Fig 5g the authors need to include the effect of CNO on control animals to show it did not have any negative effects overall (regardless of if they had a transplant or not).

4. The lack of effect of the cortical transplants was interesting but the authors do not show correlation with lesion size for these animals in the same way as they did for the SC cultures . this needs to be done to show there is no correlation for cortex (overall the SC cultures had no effect either - as shown in Fig a and b.

5 . the RNAseq seemed like an add on and did not really add to the paper. Had there been some lack of function and gain of function experiments using genes found to be correlated with function this would have been more interesting.

Reviewer #3 (Remarks to the Author):

The authors assessed the effect of human neuroepithelial stem cells derived from the developing spinal cord (SC-NES cells) on spinal cord injury (SCI) in a mouse model. After dorsal hemisection of the thoracic spinal cord of genetically immunodeficient mice, human SC-NES cells were grafted into the mice. Grafted SC-NES cells differentiated toward a neuronal fate and elongated long-distance axons in the host spinal cord. The host corticospinal tract appeared to form synapses with the grafted cells. SC-NES cell-grafted mice showed better motor recovery, compared with mice without SC-NES cell grafts. Transient silencing of SC-NES cells led to an abrogation of the effect. Transplantation of NES cells derived from the neocortex showed no beneficial effect. Transcriptome analysis supported the observed functional effects of SC-NES cells on the SCI.

The reviewer appreciates the authors' effort to characterize the transplanted SC-NES cells in vivo. However, because the importance of the study is based on a significant beneficial effect of the SC-NES cell graft in a mouse model of SCI, more thorough in vivo analyses are necessary. One concern is that the SC-NES cell graft induced a rather small effect on voluntary movement of the hindlimbs; therefore, the authors should perform multiple tests to measure motor functions in the mice. In addition, a control experiment should be performed appropriately, as mentioned in point 4 (below). Overall, the relatively small effect of the SC-NES cell graft on SCI in this mouse model diminishes enthusiasm for this study, as numerous prior studies have demonstrated beneficial effects from the implantation of various types of cells into mouse models of SCI. Specific comments are as follows:

Major points

1. SC-NES cell-grafted mice showed better motor recovery compared with mice without SC-NES cell grafts, as assessed by BMS score (Fig. 5a). However, the difference in motor recovery was

very small. The authors should perform multiple tests to measure motor function. In addition, as the difference was found only at 7 and 8 weeks, measurement of the BMS score should be performed beyond 8 weeks.

2. Fig. 5b: The percentage of spared tissue in the SC-NES cell graft group appears to be reduced, although not significantly. Did the authors perform a power analysis to exclude type 2 error?

3. The authors should also employ the contusion model of mouse spinal cord injury, as this is a clinically relevant model. The result may provide practical information regarding the development of cell transplantation therapy. As the effect of the SC-NES cell graft in this hemisection model is mild, assessment of its effect in a clinically relevant model is mandatory.

4. The control mice received a vehicle injection. The authors should provide other controls, such as grafts of other types of cells, in at least one portion of the experiment.

5. The percentage of spared tissue in each group is variable (Fig. 5c), suggesting that the hemisection was performed inappropriately.

We thank the Reviewers for recognizing the high impact of these studies. We have revised the manuscript to address each Reviewer concern, according to the plan discussed with our Editor. We have added a new Fig. 5f, new Suppl. Figs. 1i, 2e, 7, 8a, 8b, 8c, 8d, 8e, and new Suppl. Table 2. The new information includes quantitation of cell differentiation, illustration of ectopic colonies, WGCNA analysis of RNAseq data and rat contusion studies.

Reviewer #1 (Remarks to the Author)

This manuscript reports on grafting human neuroepithelial stem cells (NEC) derived from either fetal spinal cord (SC-NEC) or fetal cerebral neocortex (NCX-NEC) and propagated and expanded in vitro and then grafted into spinal cord injury (SCI) lesions of immunodeficient (NOD/SCID) mice. The NEC were characterized extensively in vitro. The SCI lesions were dorsal hemisections, a form of incomplete SCI. The NEC suspended in medium plus BDNF were grafted into lesion sites 10 days after SCI. The main findings reported are: (1) Human SC-NES are self-renewing in vitro and can be propagated and expanded in vitro for long periods. SC-NES can be differentiated to neurons and glia in vitro. (2) Human SC-NES survive well after grafting into immunodeficient mice and differentiate into propriospinal neurons that send axons into host tissue. (3) SC-NES derived neurons form contacts with host neurons. (4) SC-NES grafted mice exhibit some locomotor improvements after SCI. (5) NCX-NEC derived cells do not efficiently form neurons or improve function when grafted into spinal cord lesions but do form neurons when grafted into cortex. (6) Transcriptome screening begins to identify molecular characteristics that may underlie specifications of neurons derived from NEC from different regions.

Technical aspects of the study appear well conducted and appropriately controlled. The data presented look to be of high quality. The microscopic images are consistent with the associated quantitative data. The data presented convincingly support the interpretations made in the text.

Overall, I found this to be a high quality study that provides useful and important new information about grafting human NEC in the context of SCI. In particular, the study provides clear evidence that regional specificity of NEC progenitor populations is critical in determining integration and function of graft-derived neurons and begins to identify relevant molecular characteristics.

Nevertheless, I have two important and not difficult to deal with points that the authors should attend to (and one minor point).

Specific comments:

1. Previous studies using fetal derived cells for grafting into spinal cord led to the problem of seeding distant ectopic colonies of grafted cells along the spinal canal (which provides a peri-ependymal stem cell niche that can attract grafted cells). Some colonies were observed as far away as the 4th ventricle, and some along the surface of the spinal cord and brain. This observation led to some controversy and there have been a number of papers commenting on this. In the last few sentences of the discussion, the authors mention previous observations, and then briefly comment that they too observed such seeding of ectopic colonies of grafted cells but provide no real detail. This topic deserves a separate (but short) section at the end of the results, including some data and possibly an image in a supplementary. As regards data, the authors should document what proportion of animals exhibited ectopic colonies (2 of 10 or whatever) and how many colonies were observed per animal and where those were located (2mm away from graft along central canal or cord surface or whatever), and how far away from the graft sites was routinely examined. This type of information is important for the field to progress and should not be glossed over.

REPLY: We agree that describing in more detail the location of ectopic clusters is important. We have added a section in the second paragraph of the Results and illustrated examples in Suppl. Figs. 2e, 2f, 3c. As now described in the main text, the vast majority of ectopic nodules were located just below meninges and were typically formed by a few cells. The number of subpial clusters varied from 1 to 10 per animal. Ectopic nodules located in the central canal were

observed less frequently (1-3 clusters per animal). In our cohort, migration through both meninges or central canal did not extend more than 12 mm from the injection site in any animal.

2. A major part of the rationale for grafting neuronal progenitors from embryonic spinal cord rather than progenitors from some other CNS region after SCI derives from the notion that spinal cord progenitors give rise to propriospinal neurons, and that propriospinal neurons can form functionally meaningful relay circuits that can convey voluntary motor control across the injury site. To do so these neurons would need to receive inputs from descending projections from the brain and relay information conveying voluntary control to motor centers below the injury. The present authors seem to assume that this is a self-evident or long-standing concept, but it is not. As recently as 15 years ago, the notion intrinsic spinal cord neurons (propriospinal neurons) could create such a relay would have been scoffed at by motor physiologists, and consequently, transplantation studies aimed at grafting neuronal progenitors were viewed with great skepticism at that time and were not vigorously studied. This view was changed by two landmark studies demonstrating that after SCI, descending brain projections form new connections with propriospinal neurons and that such connections can relay information. Bareyre et al (2004) showed that after partial SCI, descending projections from the brain can form new connections with propriospinal neurons and form new relay circuits that bypass the incomplete injuries. Courtine et al (2008) showed that after multiple staggered SCI lesions that completely remove all direct descending projections from the brain, the propriospinal network of neurons can form new relay circuits that are required and sufficient to convey voluntary locomotor commands from the brain to locomotor centers below the injuries. Without these studies, the type of grafting reported here would still be widely viewed with skepticism. Indeed, the terminology and concept of 'relay circuits' after SCI, which the current authors use in their title and discussions, was originated and validated by these studies. The present authors should cite these two studies and state that these studies underpin the rationale for transplanting NES that can form spinal cord intrinsic propriospinal neurons that have the potential to form functionally meaningful relay circuits after SCI. (The papers referred to in references 45 and 46 did not originate or validate this concept). This can only strengthen the notion that grafting regionally specific neuronal progenitors is important to achieve functional benefit as indicated by the authors results here.

Bareyre et al (2004) The injured spinal cord spontaneously forms a new intraspinal circuit in adult rats. *Nat Neurosci* 7:269-277.

Courtine et al (2008) Recovery of supraspinal control of stepping via indirect propriospinal relay connections after spinal cord injury. *Nature Medicine* 14:69-74.

REPLY: We agree with the reviewer's comment and have rephrased the sentence as well as updating the bibliography to include the two above-mentioned publications.

3. Minor - The abstract should mention that the host animal for human NES transplantation in vivo was immunodeficient (NOD/SCID) mice (not just 'rodent').

REPLY: We have modified the text accordingly.

Reviewer #2 (Remarks to the Author):

This is an interesting paper where the authors have isolated either spinal cord or cortical tissues from very young human embryos, expanded them in culture and then transplanted them in to a mouse model of SCI. The conclusions of this paper are that the expanded cells from the SC but not the Ctx can restore some function in this model. There is a correlation with specific gene sets. The implications are that they have a new potential cell therapy for SC in humans, with the caveat that some animals seemed to retain some rosettes and had some aberrant migration of the cells. Overall this was an interesting study from a very well-established investigator. However, there have been many similar studies previously with very

similar results questioning the novelty of the findings (e.g. Kadoya et al, Nat Med, 2016). Furthermore, there are serious methodological and interpretational shortcomings that dampen enthusiasm for this report.

REPLY: We cite and acknowledge the investigation performed by Kadoya and coworkers. However, we wish to emphasize that, unlike the previous publication, our study of graft positional identity has utilized cells exclusively of human origin. Moreover, while the work of Kadoya et al. described the effect of graft identity on the degree of host corticospinal tract regeneration, we demonstrate a substantial difference in the extent of anatomical integration by the grafted cells, and of molecular differentiation by the graft using global transcriptome analysis. Thus, our findings provide distinct new information regarding the regional identity determinants of human cell engraftment into the spinal cord.

Major Concerns:

1 . The authors need to show more details of their expansion of the NSC's to support the conclusion that they are stable (giving rise to similar numbers of neurons) and karyotypically normal at every passage. At minimum they should show expansion rates over time, sequential plate downs and counts of neurons/astrocytes produced, and at least three karyotypes at the start, middle (6 months) and end of their expansions. In addition the metaphase spread in Fig 1g looked squashed and it was difficult to read.

REPLY: According to the reviewer's suggestion, we have provided additional details about the cell passage in the Methods section of the manuscript. There was no observable change in expansion rate with passage number. Moreover, we have provided a histogram in Suppl. Fig. 1i illustrating the percentage of neurons and glial cells obtained upon *in vitro* differentiation.

The karyotype analysis that we present in Fig. 1 was performed at passage 25, one of the highest passages reached *in vitro*. It is appropriate to assume that if the cells were euploid at this stage, then they were euploid at earlier passages. Obtaining karyotypes at greater than 25 passages (>6 months culture time) is not practical within a reasonable time frame, but the 25 passage data demonstrate robust chromosomal stability. We also re-checked the quality of the metaphase spread presented in Fig. 1g and confirm that the image in the panel faithfully reproduces the original raw data.

2. The authors did the best job they could with what was very weak functional data . This topic has been extensively discussed by NIH due to very variable results in the field - even to the point of generating testing centers for SC injury models and recovery (Experimental Neurology , Volume 233, Issue 2, February 2012, Pages 597- Replication and reproducibility in spinal cord injury research) The only significant effect came when they looked at the smallest lesions . This type of post hoc analysis while valid - speaks to the weakness of the data. It would be far more convincing to repeat the study focusing on producing milder lesions and establishing differences between transplanted and non transplanted groups. Post hoc decisions about cut off points (in this case 25%) can be a biased procedure based on particular experiments.

REPLY: The existing data are presented in a clear and statistically valid manner that documents all animals studied in Fig. 5a, 5b and 5c, with a delineated post-hoc classification in 5d. Importantly, Fig. 5c shows a significant difference between the SC-NES and Control group across all mice. This difference is not detected when comparing NCX-NES and Control groups (Suppl. Fig. 5i).

Additionally, in order to strengthen our *in vivo* investigation (as also suggested by Reviewer #3) we report the outcome of spinal cord-derived NSC transplantation in a contusion model (new Suppl. Fig. 8).

We also wish to emphasize that the behavioral data should be assessed in context. The primary focus and conclusion of our study is that cell transplants into spinal cord contusion or hemisection sites produce extensive engraftment with graft axon outgrowth. Despite this marked anatomical success, the functional recovery is relatively limited in these models. This is consistent with certain published data with other cell transplants. For example, a study from Lu and Tuszynski (Neuron, 2014, 83:789-796) showed no functional recovery despite extensive engraftment. It is an experimental conclusion, rather than technical limitation, that the behavioral result is “weak” in comparison to the “robust” anatomical results. For the majority of our studies, the anatomical result is paramount to behavior and the histological analysis is presented in detail. Behavioral function is the primary outcome only in the CNO experiment, and there we show 4 measures: BMS, grid walking, stride length and base of support.

3. The CNO experiment was a heroic attempt to show that connectivity was important. However, figures 5e and f were not convincing. There was certainly a trend down and up upon CNO administration suggesting an effect on function, but the control animals seemed to be extremely variable calling into question the reliability of the BMS score at 8 weeks. Combined with the weak overall effects of the transplant this further dampened enthusiasm. In Fig 5g the authors need to include the effect of CNO on control animals to show it did not have any negative effects overall (regardless of if they had a transplant or not).

REPLY: To clarify the effects of CNO on an animal by animal basis, we created the new graph in Fig. 5f, which clearly illustrates the decrease in the BMS score produced by the CNO administration in the cell-treated group compared to controls.

The requested CNO controls are in Suppl. Fig. 4e and 4f.

4. The lack of effect of the cortical transplants was interesting but the authors do not show correlation with lesion size for these animals in the same way as they did for the SC cultures. This needs to be done to show there is no correlation for cortex (overall the SC cultures had no effect either - as shown in Fig a and b).

REPLY: The NCX-NES control plots are in Suppl. Fig. 5i.

5. The RNAseq seemed like an add on and did not really add to the paper. Had there been some lack of function and gain of function experiments using genes found to be correlated with function this would have been more interesting.

REPLY: We believe these data provide a critical dimension and will serve as the basis for future study by ourselves and others. The data set will be publicly available.

In order to strengthen our analysis and extract additional biological information, we applied weighted gene co-expression analysis (WGCNA) to identify gene modules with similar variation across *in vitro* and *in vivo* samples (new Suppl. Fig 7 and new Suppl. Table 2). The WGCNA analysis highlights the upregulation of genes related to neurogenesis, axonogenesis and synaptic transmission in grafted SC-NES compared to other groups.

Reviewer #3 (Remarks to the Author):

The authors assessed the effect of human neuroepithelial stem cells derived from the developing spinal cord (SC-NES cells) on spinal cord injury (SCI) in a mouse model. After dorsal hemisection of the thoracic spinal cord of genetically immunodeficient mice, human SC-NES cells were grafted into the mice. Grafted SC-NES cells differentiated toward a neuronal fate and elongated long-distance axons in the host spinal cord. The host corticospinal tract appeared to form synapses with the grafted cells. SC-NES cell-grafted mice showed better motor recovery, compared with mice without SC-NES cell grafts.

Transient silencing of SC-NES cells led to an abrogation of the effect. Transplantation of NES cells derived from the neocortex showed no beneficial effect. Transcriptome analysis supported the observed functional effects of SC-NES cells on the SCI.

The reviewer appreciates the authors' effort to characterize the transplanted SC-NES cells *in vivo*. However, because the importance of the study is based on a significant beneficial effect of the SC-NES cell graft in a mouse model of SCI, more thorough *in vivo* analyses are necessary. One concern is that the SC-NES cell graft induced a rather small effect on voluntary movement of the hindlimbs; therefore, the authors should perform multiple tests to measure motor functions in the mice. In addition, a control experiment should be performed appropriately, as mentioned in point 4 (below). Overall, the relatively small effect of the SC-NES cell graft on SCI in this mouse model diminishes enthusiasm for this study, as numerous prior studies have demonstrated beneficial effects from the implantation of various types of cells into mouse models of SCI. Specific comments are as follows:

Major points

1. SC-NES cell-grafted mice showed better motor recovery compared with mice without SC-NES cell grafts, as assessed by BMS score (Fig. 5a). However, the difference in motor recovery was very small. The authors should perform multiple tests to measure motor function. In addition, as the difference was found only at 7 and 8 weeks, measurement of the BMS score should be performed beyond 8 weeks.

REPLY: Please see the response to Reviewer 2, point 2 above.

For the majority of our studies, the anatomical result is paramount to behavior and the histological analysis is presented in detail. Behavioral function is the primary outcome only in the CNO experiment, and there we show 4 measures: BMS, grid walking, stride length and base of support.

Additionally, in order to strengthen our *in vivo* investigation, we have added the outcome of spinal cord-derived NSC transplantation in a contusion model (new Suppl. Fig. 8).

2. Fig. 5b: The percentage of spared tissue in the SC-NES cell graft group appears to be reduced, although not significantly. Did the authors perform a power analysis to exclude type 2 error?

REPLY: The difference in tissue sparing is non-significant. Statistical analysis of the sample numbers and standard deviation show 80% power to detect a difference $\geq 22\%$ with $P < 0.05$. The experiment design cannot exclude smaller differences.

3. The authors should also employ the contusion model of mouse spinal cord injury, as this is a clinically relevant model. The result may provide practical information regarding the development of cell transplantation therapy. As the effect of the SC-NES cell graft in this hemisection model is mild, assessment of its effect in a clinically relevant model is mandatory.

REPLY: As suggested by the reviewer, we supported our investigation by adding data from a spinal cord contusion model (new Suppl. Fig. 8). The contusion was performed in rats because the procedure is more reproducible in the larger species, and we used cyclosporine to achieve adequate immunosuppression in place of genetic immunodeficiency. For the hemisection studies, we observed a lack of myelination of human axonal fibers, which might contribute to the limited behavioral benefit of SC-NES cells. Therefore, for the contusion study, we performed an allograft preparing NSCs from the spinal cord primordia of rat embryos. Our histological data show that, similarly to the hemisection model, SC-derived NSCs integrate into the host spinal cord and extend many axons into host tissue. Additionally, the implantation of the cells induces a significant, though still limited, amelioration of the hindlimb motor function of recipient animals by BBB score.

Of note, we detected no rifts in the contused spinal cords. The rifts found in the hemisectioned cords were likely due to an infiltration of meningeal cells in the spinal cord parenchyma as a consequence of the hemisection surgical procedure. Since in the contusion the

meninges are left intact, the graft is not segregated into two separated components and this may explain the anatomical result.

4. The control mice received a vehicle injection. The authors should provide other controls, such as grafts of other types of cells, in at least one portion of the experiment.

REPLY: We have provided controls using other cells. Specifically, the NCX-NES cells provide a directly parallel experiment with an age-matched cell line that is nearly identical *in vitro* but has different regional identity and yields a very different outcome *in vivo* (Fig. 6).

5. The percentage of spared tissue in each group is variable (Fig. 5c), suggesting that the hemisection was performed inappropriately.

REPLY: The correlation analysis in Fig. 5c shows that effect of SC-NES is significant across lesion sizes. The variability of depth may be larger than is typical in such experiments because mice must undergo a second transplant surgery after hemisection.

REVIEWERS' COMMENTS:

Reviewer #1 (Remarks to the Author):

In this revised manuscript the authors have dealt with my concerns, and in my opinion have appropriately responded to the requests of the other reviewers. I still find that the paper is technically strong and makes a important contribution to the field of stem cell grafting after SCI.

Reviewer #2 (Remarks to the Author):

The authors have addressed my technical concerns and the inclusion of the new model is good although again with only small functional gains relative to the anatomy. In response they do concede that the functional gains are small (as highlighted by another reviewer) and in this reviewers opinion the study could be considered incremental when compared to published studies.

Reviewer #3 (Remarks to the Author):

The authors have now addressed many of the issues previously raised by the reviewers, in the manuscript as well as in the direct response to the comments. The results for additional experiments have been provided to strengthen some of the claims made in the previous versions of this manuscript. However, I still judge that the relatively small effect of the SC-NES cell graft on SCI in this mouse model diminishes enthusiasm for this study, as numerous prior studies have demonstrated beneficial effects from the implantation of various types of cells into mouse models of SCI. The data in the revised version contribute to relatively little conceptual advances in the field.